# A new precipitation emulator (PREMU v1.0) for lower complexity models

Gang Liu[1], Shushi Peng[1*], Chris Huntingford[2], Yi Xi[1]

[1]Sino-French Institute for Earth System Science, College of Urban and Environmental Sciences, Peking University, Beijing, China
[2]UK Centre for Ecology and Hydrology, Wallingford, Oxfordshire OX10 8BB, United Kingdom

*Correspondence to*: Shushi Peng (speng@pku.edu.cn)

**Abstract.** Precipitation is a crucial component of the global water cycle. Rainfall features(e.g., strength or frequency)strongly affect societal activities and are closely associated with the functioning of terrestrial ecosystems. Hence predicting global and gridded precipitation under different emission scenarios is an essential output of climate change research, enabling a better understanding of future interactions between land biomes and climate change. Some current lower complexity models (LCMs) are designed to emulate precipitation in a computationally effective way. However, for precipitation in particular, they are known to have large errors, due to their simpler linear scaling of precipitation changes against global warming (e.g., IMOGEN; Zelazowski et al., 2018). Here, to reduce the errors in emulating precipitation, we provide a data-calibrated precipitation emulator (PREMU), offering a convenient and computationally effective way to estimate and represent precipitation well, as simulated by different Earth system models (ESMs) and under different user-prescribed emission scenarios. We construct the relationship between global/local precipitation and modes of global gridded temperature and find that the emulator shows a good performance in predicting historically observed precipitation from GSWP3. The ESM-specific emulator also estimates well the simulated precipitation of nine ESMs and under four dissimilar future scenarios of future atmospheric greenhouse gases (GHGs). Our ESM-specific emulator also reproduced well interannual fluctuations (R = 0.82–0.93, p < 0.001) of global land average precipitation (GLAP) simulated by the nine ESMs, as well as their trends and spatial patterns. The default configuration of our emulator only requires gridded temperature, also available from lower complexity models such as IMOGEN (Zelazowski et al., 2018) and MESMER (Beusch et al., 2022; Nath et al., 2022), which themselves are calibrated against ESMs. Therefore, our precipitation emulator can be directly coupled within other LCMs, for instance improving on the current emulations of precipitation implicit in IMOGEN. The PREMU model has the opportunity to provide the driving conditions to model well the hydrological cycle, ecological processes, and their interactions with climate change. Critically the efficiency of LCMs allows them to make projections for many more potential future trajectories in atmospheric GHG concentrations than is possible with full ESMs, due to the high computational requirement of the latter. By coupling with PREMU, LCMs will have the ability to emulate gridded precipitation, thus they can be widely coupled with hydrological models or land surface models.

# 1 Introduction

Earth system models (ESMs) are the primary tools to study the impact of greenhouse gas emissions on our climate, representing all the important Earth system processes. (IPCC, 2013). However, there is a lack of sufficient computational power to run the most comprehensive, physically complete climate models for every application of interest (Nicholls et al.,2020), or for every potential future emissions scenario. Thus, Lower complexity models (LCMs) are designed as the common approaches to improve computational efficiency in climate change research by focusing on the most impact relevant variables (Gasser et al., 2017). By describing highly parameterized properties of the climate system, LCMs are many orders of magnitude faster than full complexity ESMs (Nicholls et al., 2020). The simplest LCMs are Energy Balance Models (EBMs) that use multiple parameterized numerical models to estimate the changes in greenhouse gas concentrations, radiative forcing and then global land/ocean average temperature under different emission scenarios (Meinshausen et al., 2011; Nicholls et al., 2020). However, such global "box" models do not simulate the spatial pattern of temperature. Some more complex LCMs, such as IMOGEN and OSCAR, additionally emulate the spatial pattern of temperature through the pattern-scaling method. Pattern-scaling multiplies global temperature change by spatial patterns to give regional information (Zelazowski et al., 2018; Gasser et al., 2017; Huntingford et al., 2010; Tebaldi and Arblaster, 2014; Tebaldi and Knutti, 2018). Joint temperature and precipitation emulation by considering anthropogenic GHG forcing and large-scale modes of Sea Surface Temperature (SST) variability has been proven possible (Mckinon and Deser 2018; Mckinon and Deser 2021). More recently, a spatially resolved emulator (MESMER) solely requiring GMT e.g. by coupling to the emission-driven LCM (MAGICC), to then generate annual temperature fields has been developed (Beusch et al., 2022). Other than temperature, however, it is still a challenge for LCMs to simulate well other climate variables such as precipitation under different emission scenarios (Gasser et al., 2017).

Precipitation has high spatio-temporal variability and is affected by atmospheric dynamics and inter-annual modes of variability (Li et al., 2021; Tsanis and Tapoglou, 2019), making representing of it within tradition al LCM approaches difficult. Thus, only two LCMs (IMOGEN and OSCAR) have tried to emulate precipitation, but with poor skill (Zelazowski et al., 2018; Gasser et al., 2017). IMOGEN emulates the gridded precipitation based on the regression relationship (by month and location) between gridded precipitation and global land average temperature (Zelazowski et al., 2018). OSCAR constructs the emulator by establishing a relationship between global average precipitation and global average temperature and radiative forcing, from which a pattern-scaling method is used to deduce the gridded precipitation (Gasser et al., 2017). Nevertheless, the gridded precipitation estimated by the simple linear method is not fully reliable in either IMOGEN or OSCAR; the gridded precipitation predicted by IMOGEN explains less than 20% of variance of seasonal precipitation in most regions (Zelazowski et al., 2018) and OSCAR cannot capture the interannual variations of regional precipitation across the globe at all (Gasser et al., 2017). This may be because 1) the global average temperature could not fully characterize local temperature and moisture recycling (Prein et al., 2017) and large-scale circulation (Shepherd, 2014; Fereday et al., 2018; Heinze-Deml et al., 2021); 2) there is at any given location a nonlinear relationship between local rainfall features and global warming (Allen and Ingram., 2002;

Collins et al., 2013; Chadwick and Good, 2013). Hence models such as IMOGEN, that rely on linear pattern-scaling, by definition cannot fully capture expected future precipitation changes. Nevertheless, precipitation is a crucial component of the water cycle (Eltahir and Bras, 1996; Trenberth et al., 2003; Sun et al., 2018), has key societal implications and is closely associated with the functioning of terrestrial ecosystems. Accurately emulating precipitation change is necessary to determine the climate response to different emission scenarios and to understand feedbacks to global warming via rainfall-dependent vegetation net primary productivity (Gasser et al., 2017).

Previous studies have confirmed the important impact of ocean-atmosphere oscillations on inter-annual and inter-decadal variations of regional precipitation (e.g., Li et al., 2021; Tsanis and Tapoglou, 2019; Dai, 2013). As such, our noted changes in gridded surface air temperature likely additionally contain information about sea surface temperature and ocean-atmosphere oscillations, as additional information on the background global land average temperature. To this end, we proposed a computationally efficient precipitation emulator (PREMU), which uses gridded temperature data as independent forcing variables to reconstruct the gridded precipitation. We have designed the presented emulator such that it can act as, for instance, an enhanced precipitation module for the OSCAR and IMOGEN models. Alternatively, PREMU could be coupled directly with the MAGICC-MESMER and MESMER-M emulators. When coupling with other LCMs, tracing gridded precipitation under novel trajectories for GHGs will be possible. In Sect. 2 and Sect. 3 we described the datasets (both measurements and ESM-based) and the methods to construct the driving-data specific emulator for historical observation and each ESM in detail. We illustrated the emulator's ability to emulate the historical and future precipitation in Sect. 4. Finally, Sect. 5 discusses the strengths and caveats of our emulator.

## 2 Data

In the analysis, we tested the performance of PREMU for both historical periods (1901-2016) and future periods (2015-2100). For historical periods, we used different time periods of observation data for calibration (1901-1950) and validation (1951-2016). While for the future, we used ESM data under different emission scenarios for calibration (SSP5-8.5) and validation (SSP1-2.6, SSP2-4.5 and SSP3-7.0).

### 2.1 Observation datasets

To verify the ability to predict historical precipitation using the emulator proposed in this study, we first demonstrated the applicability of the emulator on observational data provided by the Global Soil Wetness Project Phase 3 (GSWP3) (Kim, 2017a). The GSWP3 dataset is based on the 20th Century Reanalysis (20CR) version 2c (Compo et al., 2011), and has precipitation fields at a resolution of 2° × 2°. The GSWP3 data is dynamically downscaled using spectral nudging and bias-correction from the Global Precipitation Climatology Project and Climate Research Unit (Humphrey and Gudmundsson, 2019). This approach successfully keeps the low frequency signal of the two original reanalysis products, yet provides additional high

frequency signals that were lacking in previous products but are essential for investigating extreme events (Kim, 2017a). GSWP3 also provides the other seven climate variables including 2-m air temperature (Tair), specific humidity, surface down welling long-wave radiation, surface down welling short-wave radiation, surface air pressure, and near-surface wind speed at a 0.5° × 0.5° spatial resolution from years 1901 to 2016. Here, with an emphasis on precipitation, we used Tair and precipitation from GSWP3 from period 1901 to 1950 to calibrate the emulator and from 1951 to 2016 to verify the emulator's

performance in predicting historical precipitation.

## 2.2 Earth System Model data

To evaluate the emulator's performance in estimating future precipitation, we additionally used monthly precipitation and 2-m Tair for years 2015 to 2100 from nine ESMs operated under four Shared Socioeconomic Pathways (SSPs; SSP1-2.6, SSP2-4.5, SSP3-7.0, and SSP5-8.5; O'Neill et al., 2016). These ESMs are included in the 6th phase of the Coupled Model

Intercomparison Project (CMIP6; Table 1). The SSP5-8.5 scenario represents the largest future change of GHG concentrations, compared to the other three SSPs (Hausfather and Peters, 2020), and so is associated with the largest variation in precipitation and Tair changes. Hence, we used the precipitation and Tair from SSP5-8.5 to calibrate the emulator and the other three scenarios to verify the emulator's performance at reproducing ESMs. But note that the emulator based on this extreme warming scenario (SSP5-8.5) may not produce well the precipitation patterns of cooler scenarios (e.g., SSP1-2.6) due to the nonlinearly

response of the atmosphere to warming. We discussed this further in Sect. 5. In addition, in view of the different responses of precipitation to the changes of Tair in different ESMs, we constructed the emulator for each ESM respectively. For comparison between different models, the precipitation and Tair from all nine ESMs are re-gridded to the resolution of 2.5° × 2.5° using the first-order conservative remapping technique (Jones, 1999). As there remains substantial error and uncertainty in gridded precipitation data, we retained only a first-order re-gridding method for precipitation or Tair (we note second order calculations

have been used by others [Brunner et al., 2020]).

## 3 Methods

### 3.1 General Approach

Given the relatively poor linear relationship between global/gridded (i.e. local) precipitation and global land mean temperature (as noted in Zelazowski et al., 2018), we try to build a new emulator for precipitation. We still search for a relationship between

precipitation change and some function of gridded Tair (Fig. 1), although more complex than simply linear in global temperature. Since precipitation is also not only controlled by local temperature (Zhou et al., 2020), we seek links to features of Tair from all grid cells. There are 10,368 gridboxes considered at a common 2.5° × 2.5° resolution, that we used to for both historical observations and future ESMs, after remapping the latter. It's this Tair data and model simulations that we used across the globe to predict precipitation variation.

In this section, we set out underlying methods for our emulation of local precipitation by the principal component regression (PCR) approach, based on relating it to the dominant modes of variability of temperature. PCR is a type of regression analysis, which considers the orthogonal principal components as independent variables. As a method that can be used to overcome the problem of multilinearity on predictor variables, the PCR technique is widely used in forecasting seasonal precipitation (Kim et al.,2017b) and reconstructing the climatic modes of variability (Jones et al., 2009; Michel et al., 2020). These authors find PCR method perform remarkably well, and the method is explained in detail in Chapter 13 of Storch and Zwiers (2011). As might be expected, therefore, our starting point is deriving the principal components of global gridded temperature. We followed the standard procedure of Principal Component Analysis (PCA), as used in climate research (e.g.,Yan et al., 2020; Singh, 2006; Jiang et al., 2020), that consist of a set of time-invariant spatial patterns multiplied by timeseries of coefficients. PCA notation can vary between users, but for simplicity, we will simply refer to these as PCA "spatial patterns" (each of which has the dimensions of the spatial grid, or alternatively an array of one dimension with 10,368 points), and PCA "timeseries", which multiply the patterns. These timeseries, derived for each month, contain values applicable for each year, and so for instance, fitting to years 1901-1950 imply they will have 50 numbers. As precipitation may be influenced by the temperature of the previous months via large scale circulation (e.g., the lag effects of ENSO on regional precipitation; Li et al., 2011; Lu et al., 2019; Efthymiadis et al., 2007), we used the average Tair of three months (month of interest plus the previous two months) to apply PCA. In addition, considering that ESMs may under-represent the effects of topography and aerosols on modelled precipitation (Samset et al., 2016; Yang et al., 2021), we calibrated the emulator for the historical period and future separately. For the estimates of future change, we used precipitation and Tair from SSP5-8.5 scenario of greenhouse gas increases to calibrate the ESM-specific emulator. We selected this scenario due to it representing the most extreme changes in Tair amongst SSPs (O'Neill et al., 2016).

## 3.2 Framework for PREMU

### 3.2.1 Calibration

We set out in Table 2 below our notation for indices and variable quantities.

Our use of the standard definition of PCAs (sum of patterns × timeseries) implies that the estimate of the principal components is given by the spatial pattern of principal component coefficients $U_{e,m}^{Spatial}(k,i)$ multiplied by the timeseries of three months average gridded temperature. The choice of the 3-month average Tair as independent variables is robust with further details illustrated in Sect. 5. The PCA component coefficients, $U_{e,m}^{Spatial}(k,i)$, are the combination of eigenvectors of the covariance matrix of the anomalies of the gridded 3-month average Tair. However, unlike many limited-area applications that derive a single timeseries (to multiply each PCA; Rahaman et al., 2019), we instead determined a timeseries for each location, and hence $T_{e,m}^{Timeseries}(k,y)$ has a $k$ dependency. We then summed over all locations $k$, and this gives a global annual timeseries (for each ESM $e$, month $m$, year $y$ and PCA $i$) of:

$$T_{e,m}^{PCA}(y,i) = \sum_{k=1}^{N_{grid}} U_{e,m}^{Spatial}(k,i) * T_{e,m}^{Timeseries}(k,y) \qquad (1)$$

Using Tair for years 1901-1950 from the GSWP3 gridded dataset, and for period 2015-2100 from the nine ESMs forced under the SSP5-8.5 scenario, we determined the principal component coefficient matrix $U_{e,m}^{Spatial}$ for each month $m$, and ESM $e$ (or GWSP3) independently. We employed Equation (1), and the top 10 principal components ($i = 1,...,10$) were used in this study, which described more than 70% of gridded temperature information (i.e. variation) across the globe (Fig. 2 and Figs. S1-S3). The stability of the component coefficients $U_{e,m}^{Spatial}(k,i)$ of the top 10 PCs, between different scenarios, is discussed in Sect. 5. Otherwise, we found that the 15 ocean-atmosphere climate indices can be well reconstructed by multi-linear regression of the top 10 principal components $T_{e,m}^{PCA}(y, 1:10)$ (Fig. S4). This suggests that the principal components of gridded temperature likely additionally contain information about sea surface temperature and ocean-atmosphere oscillations, which have great impacts on inter-annual and inter-decadal variations of regional precipitation.

Having established the temperature principal components, via the form presented in Eq (1), we then mapped these onto precipitation both globally and locally. Our mapping is via a standard regression based on 50 datapoints (i.e. year 1901-1950; or 86 datapoints in ESMs for year 2015-2100). The regression has 40 degrees-of-freedom, due to our selection of 10 PCAs, and this relationship links the timeseries of monthly gridded precipitation from the GSWP3 dataset (or the nine ESMs) and the 10 principal components of gridded temperature for each month of the 12 months of the year (January to December). This regression (for each ESM $e$, month $m$, grid $k$ and year $y$) is constructed as:

$$P_{e,m}^{grid}(k,y) = \sum_{i=1}^{N_{PCA}} \beta_{e,m}(k,i) * T_{e,m}^{PCA}(y,i) + \beta_{e,m}^{intercept}(k,i) \qquad (2)$$

where $P_{e,m}^{grid}(k,y)$ is the precipitation at a specific cell, $k$. Variable $\beta_{e,m}$ represents the regression coefficient of each principal component and $\beta_{e,m}^{intercept}$ is the intercept term, which are derived from linear regressions by least squares method using the calibration time series. To provide the reliable estimation of global monthly precipitation $P_{e,m}^{\widehat{global}}(y)$, we constructed the regression relationship between GLAP and the 10 individual principal components separately, following Eq. (2).

### 3.2.2 Generating emulations using PREMU

For validation, we set out in Table 3 below the variables that we estimated with our methodology. The "overhat" notation represents an estimated quantity. We first recalled that we fitted our PCA-based framework to historical temperature and precipitation data for period 1901-1950, and to ESMs for the SSP5-8.5 future atmospheric GHG concentration scenario. Based on the principal component coefficients extracted using these calibration datasets, we then estimated the $T_{e,m}^{\widehat{PCA}}(y, i)$ using Eq.1, for 1951-2016 using Tair from GSWP3 and for 2015-2100 using Tair from each ESM independently under the other three SSPs. Then we estimated the $P_{e,m}^{\widehat{global}}(y)$ and $P_{e,m}^{\widehat{grid}}(s,y)$ individually by the $T_{e,m}^{\widehat{PCA}}(y, i)$ and our fitted regression coefficients in Eq. 2.

### 3.3 Validation

Before evaluating the performance of PREMU, we found a slight difference in GLAP between $P_{e,m}^{\widehat{global}}$ and the spatial average

of $\widehat{P_{e,m}^{grid}}$, brought by setting negative numbers of $\widehat{P_{e,m}^{grid}}$ to be zero at some grid points. Thus, in a final component to our

calculations, we scale $\widehat{P_{e,m}^{grid}}$ by the ratio of $P_{e,m}^{\widehat{global}}$ and the monthly GLAP from average over grid points as area-averaged

$\widehat{P_{e,m}^{grid}}$, i.e. with equation:

$$P_{e,m}^{grid,\,\widehat{Adj}}(k,y) = P_{e,m}^{\widehat{grid}}(k,y) * \frac{P_{e,m}^{\widehat{global}}}{Mean\left(\widehat{P_{e,m}^{grid}}\right)}, \tag{3}$$

where $P_{e,m}^{\widehat{grid,Adj}}$ is the adjusted estimation of monthly gridded precipitation and $Mean\left(\widehat{P_{e,m}^{grid}}\right)$ represents the area-weighted

averaged $\widehat{P_{e,m}^{grid}}$.

    In order to evaluate the advantages of PREMU compared to the emulations of gridded precipitation by other emulators (e.g. IMOGEN-based or OSCAR-based method), we simply used a prediction that linearly relates rainfall changes to global

temperature changes and variability. We compared this simpler linear approach with the performance of PREMU. To evaluate how well the performance of PREMU at describing the historical observations, we compared the MAP and trends of GLAP from GSWP3 with the emulated values obtained from PREMU and the simple linear approach (subsection Sect. 4.1). Our statistic to compare was the Pearson correlation coefficients between the GLAP from observations and these two emulations. For gridded precipitation, we instead calculated the percentage error of MAP during 1987-2016 and compared the changes of

MAP in the first and last three decades of the validation period (1951-1980 and 1987-2016) for each grid. Similarly, we evaluated the performance of PREMU at describing future precipitation by comparing the MAP, and trends of GLAP with ESMs, for the four future scenarios (subsection Sect. 4.2). As PREMU is an ESM-specific emulator, we calculated the performance of PREMU on each ESM individually. The percentage error at each grid point was used to evaluate the emulations of PREMU at different locations. We additionally evaluated the performance of PREMU on the seasonal cycle of precipitation

(subsections Sect. 4.3.) Here, we compared the land average precipitation in different latitudes from GSWP3 or multi-model mean of 9 ESMs with the emulation of PREMU. Also, the error in spatial pattern of seasonal precipitation is used to show that PREMU can capture the seasonal cycle of gridded precipitation from both historical observations or ESMs.

    Furthermore, we evaluated the reliability of our assumption of constant spatial part of PCAs in this study in Sect. 5. To test

this assumption, we compared the coefficient matric of temperature derived for the SSP5-8.5 scenario with those from the SSP1-2.6 scenario. Finally, for the further developments of PREMU, we explored the performance of emulating precipitation by other versions of PREMU (e.g., PREMU-mon, PREMU-6mon and PREMU-land), and again our findings are presented in Sect. 5.

## 4 Results

### 4.1 Performance of precipitation emulator on historical precipitation

As outlined above, to evaluate the ability of our PREMU to emulate historical precipitation, we first used the precipitation and Tair data from GSWP3, for the period 1901-1951, to calibrate its parameters. We then tested its predictive capability at estimating precipitation data from the period 1951-2016. For the calibration period (1901-1950), our emulator shows a consistent areally-averaged global annual precipitation value (1002 mm year$^{-1}$ and 1002 mm year$^{-1}$ for PREMU and GSWP3, respectively) and trend (0.48 mm year$^{-2}$ and 0.54 mm year$^{-2}$ for PREMU and GSWP3; Table S1). Furthermore, the interannual variations of GLAP derived from PREMU are found to be significantly correlated with that from GSWP3 (R = 0.81, p < 0.001). For the validation period (1951-2016), PREMU also captured the observed trend (0.22 mm year$^{-2}$ for both PREMU and GSWP3) and interannual variations (R = 0.67, p < 0.001) of GLAP from GSWP3 (Fig.3a). By contrast, if we simply used a prediction that linearly relates rainfall changes to global temperature changes and variability, which has similarities to the IMOGEN-based method, we found a much smaller (-0.25 mm year$^{-2}$, -45%) and larger (+0.47 mm year$^{-2}$, +213%) trend in GLAP than GSWP3 for the calibration and validation periods, respectively. In addition, the interannual variations of GLAP as estimated by a simple linear regression against global temperature, shows a weaker correlation with GSWP3 in these two periods (R = 0.27, p = 0.06 for the calibration period; R = 0.15, p = 0.21 for validation period; Fig. 3a; Table S1).

Spatially, PREMU and a simple linear-based method (similar to the algorithms in IMOGEN) can both emulate similarly the spatial pattern of mean annual precipitation (MAP) during the last three decades in the validation period (1987–2016), as observed in the GSWP3 data (Fig. 4a, c, e). Compared to the simple linear approach, there are fewer grid cells showing more than 25% error of MAP from GSWP3 in our emulation (17%; Table S1) against a simple linear fit (27%; Fig. 3b,c). The overestimation of MAP with our PREMU method is mainly found in the Tibetan Plateau and central Africa (~20%), and the underestimation of MAP mainly in northern Siberia, Greenland Island and northern Australia (-15% – -30%; Fig. 3c and Fig. S5). To verify the emulator's ability to predict changes in gridded observation precipitation, we calculated the change of MAP in the first and last three decades of the validation period (1951-1980 and 1987-2016; Fig. 4b, d, f). For the differences between these two time periods, PREMU shows consistent changes of precipitation in northern Eurasia, North America and the central South America and when using GSWP3 data. However, the simple linear method has underestimated the increase or overestimated the decrease of precipitation in these regions (Fig. 4). In some regions of East Asia, Europe, Australia and South America, PREMU underestimates / overestimates the changes in annual precipitation (in range 50-200 mm year$^{-1}$; Fig. S6), and these values represent little improvement over the simple linear method (Fig. 4). Furthermore, we noted that the changes in MAP from both PREMU and the simple linear method shows the opposite to the changes in MAP from GSWP3 in some regions (e.g., Australia and West Africa; Fig. 4b, d, f). This suggests that changes in precipitation in these particular regions may be driven by the factors such as aerosols, topography and land use changes rather than temperature, which are further discussed in Sect. **5**.

## 4.2 Performance of precipitation emulator on future precipitation from CMIP6 ESMs

A key requirement of PREMU is that it can make projections of precipitation change for different future scenarios of atmospheric GHG concentrations, and potentially for novel trajectories of such gases for which ESMs have not made calculations. To test for this capability, we analysed its performance at predicting changes under the SSP1-2.6, SSP2-4.5 and SSP3-7.0 scenarios, and for which ESM-based simulations that are available. Recall that our PREMU calibration was against ESMs operated for the SSP5-8.5 scenario, capturing inter-ESM differences in projections of Tair, and critically, precipitation. Similar to the emulation for the historical period, PREMU shows a good performance in emulating future precipitation (Fig. 5). For the calibration scenario, the multi-model mean GLAP from PREMU shows consistent trend (0.96 mm year$^{-2}$ and 0.96 mm year$^{-2}$ for our emulator and ESMs; Table S2) and interannual variation (R = 0.98, p<0.001) with that from ESMs (Fig. 5a). For the three validation scenarios of the different SSPs, PREMU shows a better correlation in interannual variations of GLAP with ESMs than the historical period (R = 0.86, p<0.001 for SSP1-2.6; R = 0.95, p<0.001 for SSP2-4.5; R = 0.95, p<0.001 for SSP3-7.0; Fig. 5c,e,g). Although the trends of global precipitation in our emulation are close to that of ESMs across the three scenarios (Table 4), the error of trends by the PREMU increases from high to low emission scenarios (Table 4; Table S3). In addition, the standard deviations of GLAP are underestimated by 2% in SSP5-8.5 and by 43% in SSP1-2.6 (Table. S2). At the individual ESM level, PREMU can capture well both trends and interannual fluctuations of GLAP well under all scenarios for MPI-ESM1-2-LR, MIROC6, EC-Earth3 and CanESM5. For the other five models, there are biases of trends (Table 4) and/or interannual variations of GLAP (Fig. 6 and Figs. S7-S9) between our emulation and ESMs. These differences could be partly related to the substantial uncertainties and different features affecting future precipitation projections in ESMs. These factors are discussed in Sect. 5.

For the spatial pattern (Fig. 5b,d,f,h), PREMU can reproduce the projected pattern of MAP from the multi-model mean of ESMs under the calibration scenario and the three validation scenarios (Fig. 7). Compared to the historical period, the error of multi-model MAP between our emulation and ESMs is relatively smaller (~10%; Fig. 7). As for the changes in precipitation during 2015-2100, PREMU shows a consistent canonical pattern of MAP from multi-model mean of ESMs (Fig. 8), with a 50-200 mm year$^{-1}$ increase of annual precipitation in Eurasia, North America and northeastern of Africa, but a 50-200 mm years$^{-1}$ decrease in Amazon rainforest from the low to high emission scenarios. However, similar to the simulation for GLAP, the error in spatial pattern of MAP and changes in MAP between our emulation and ESMs increases from high to low emission scenarios (Fig. 5 and Fig. 8). Furthermore, compared to the historical period, PREMU can capture the changes in MAP in West Africa well under all four scenarios, but also emulated the opposite changes of MAP in Australia under SSP3-7.0, which is discussed in Sect. 5. When considering performance at the individual ESM level, PREMU can reproduce the spatial pattern of MAP well. In general, the proportions of grid cells with an error more than 10% is small (8% [6%-14%] for SSP1-2.6; 5% [3%-8%] for SSP2-4.5; 5% [3%- 9%] for SSP3-7.0; 0% [0%- 0 %] for SSP5-8.5; mean [min-max]; Fig. 9 and Figs. S10-S12). However, as for the changes in MAP, we found a relative poor performance in emulating the changes in some ESMs especially

in SSP1-2.6 (Fig. 10 and Figs. S13-S15). While the spatial pattern of changes in MAP is quite different across ESMs even under the same SSP, the errors by PREMU are much smaller than the inter-ESM differences (Fig. 10 and Figs. S13-S15).

### 4.3 Seasonal performance of precipitation emulator

As a monthly emulator, the performance of PREMU at describing the seasonal cycle of precipitation requires evaluation. For the historical observations, PREMU can capture the seasonal cycle of GLAP in each latitude band (Fig. 11). There are some

spatial differences, with little error in boreal regions but with 18-22 mm mon$^{-1}$ error in tropics (Fig. S16). We found fewer grid cells showing more than 20 mm mon$^{-1}$ error of seasonal precipitation from GSWP3 in our emulation (18% - 24%) compared to using a simple linear fit (24% - 31%; Fig. S16). As for the future precipitation, PREMU shows a good performance on emulating the monthly land average precipitation in each latitude band under all scenarios (Fig. S17). However, PERMU tends to overestimate the JJA (June, July, August) and SON (September, October, November) precipitation in South Asia and

Amazon and underestimates the SON and DJF (December, January, February) precipitation in West Africa under SSP1-2.6, while the error in spatial pattern of seasonal precipitation decreases from low to high emission scenarios (Fig. S18).

### 5 Discussion

To our knowledge, this study provides a novel approach to linking local precipitation changes to geographical features (i.e. spatial modes) of gridded temperature in a single emulator chain, which can be further incorporated into existing LCMs.

Despite relying on a series of simple assumptions, PREMU can successfully capture the simulated changes in precipitation by ESMs and under a broad range of different emission scenarios. Comparing with the precipitation predicted by a simple linear regression between local rainfall alteration and level of global warming (e.g. as used in other LCMs such as IMOGEN), the rainfall in PREMU shows more consistent trends and interannual variations of GLAP and in the spatial pattern of MAP. These improvements are noted in the comparison against the historical precipitation recorded in the GSWP3 dataset, as well as when

emulating the predicted future precipitation by ESMs (Figs. 3-6). For a user-prescribed emission scenario, or for a time-evolving global temperature trajectory designed to constrain warming to a level such as two degrees (Huntingford et al., 2017), PREMU can accurately and quickly emulate the related gridded precipitation changes. Our method can utilize the existing spatial features of gridded temperature from either ESMs or LCMs, to support studies related to future changes in precipitation. In particular, coupling our precipitation emulator with other LCMs provides the input climate forcing for land surface models

to help disentangle hydrological and ecological responses globally to future climate change (Zelazowski et al., 2018; Li et al., 2022; Korell et al., 2021).

### 5.1 Possible cause for the emulation errors of PREMU

We noted that the performance of PREMU at predicting future precipitation from ESMs is much better than that when emulating historical GSWP3 precipitation. This is because PREMU is only based on the ten modes of global gridded

temperature, and the effects of aerosols and topography on precipitation are not well represented in our emulator (Austin and Dirks., 2005; Medvigy et al., 2012). For instance, Fig. 3 shows emulated precipitation has large errors in mountainous regions, where orographic precipitation could be dominant. The effect of raised aerosols in the 20[th] century have been shown as having a role almost as large as greenhouse gases warming effects on features of regional precipitation (e.g. in India; Bollasina et al., 2011). Thus, adding the effects of topography and aerosols on precipitation in to the PREMU could further improve the ability of predicting precipitation. Considering the changes in MAP in West Africa from PREMU are opposite to the changes in MAP from GSWP3 (Fig. 4), PREMU can emulate the changes in MAP in West Africa from ESMs well. An alternative argument is that the good performance on predicting precipitation from ESMs by PREMU may suggest that climate models under-represent the effects of topography and aerosols (Samset et al., 2016; Yang et al., 2021).

Overall, the performance of predicting precipitation from ESMs by our emulator is good, although not for all models and under all scenarios. Both trends and interannual variability of GLAP are well captured under all scenarios for MPI-ESM1-2-LR, MIROC6, EC-Earth3 and CanESM5, but PREMU performs less well at predicting GLAP in the other five ESMs (Fig. 6 and Figs. S7-S9). This may be because different ESMs use alternative atmospheric circulation model and precipitation schemes (e.g., CAM6.3 in CESM2 and AM4 in GFDL-ESM4), which contain different physical processes or parameterizations in their simulation of precipitation (Danabasoglu et al., 2020; Horowitz et al., 2020; Hourdin et al., 2020). Notable is that, PREMU underestimates 30% (6%–60%) of interannual variations of precipitation in all ESMs and when considering across all four GHG concentration scenarios. This is a common "feature" of linear regression models, as they favor bias reduction over variance under the bias-variance trade-off (Geman et al.,1992). In addition, we also suggest this may be because of missing some modes (out of the 10 modes we used) for interannual variations, or an under-representation of the effect of key climate modes such as ENSO in our emulator, or a non-linear response of precipitation to climate modes. It is also worth noting that the trend of GLAP in CESM2 and GFDL-ESM4 under SSP3-7.0 are lower than that under SSP2-4.5. We speculate that this may be caused by the most drastic land use changes associated with that former SSP scenario, resulting in a slower increase in precipitation under SSP3-7.0 (Riahi et al., 2017). Our emulator predicts precipitation through the temperature gradients and so cannot capture the impact of land-use changes on precipitation via altered land-atmospheric feedbacks impacts the hydrological cycle (Table 4).

### 5.2 Evaluating the assumptions in methods

Our emulator is based on the assumption that spatial temperature modes can describe well precipitation. Hence, we assumed implicitly that the coefficient matrix in PCA for global gridded temperature (i.e., the weights of each grid cell for each principal component) are stable, i.e. invariant, with climate change. To test this assumption, we compared the coefficient matrixes of temperature derived for the SSP5-8.5 scenario with those from the SSP1-2.6 scenario, for modelled period 2015-2100 (Figs. S16 and S17). For the future scenarios, we found that most coefficients of PCA, in both January and July, are similar between

the SSP5-8.5 scenario and SSP1-2.6 scenario. Though with a different order of PCA coefficients (Fig. S19-S20), this finding suggests that the main features of global temperature are constant across different scenarios. We noted in some instances, the signs of coefficients are opposite for some principal components between different scenarios, but this corresponds to regression coefficients also with opposite signs, which combined give the same predictions. Depending on circumstance, we noted that it may not be necessary to use all top ten PCs. For instance, if researchers only require the decadal average or any increasing trend in precipitation, PREMU calibrated by the top 1 PC of Tair under SSP5-8.5 may be sufficient to capture these characteristics of GLAP from ESMs and under all four scenarios (Fig. S21). If the researchers only require the decadal average or increase trend of precipitation, PREMU calibrated by the top 1 PC of Tair under SSP5-8.5 can also capture the trends of GLAP from ESMs under all four scenarios (Fig. S21). Furthermore, we have assumed that the coefficients of PCA are linked with the ocean-atmosphere oscillations, but the detailed physical explanations of the coefficient matrix need further study in the future.

We have confirmed that our ability to reproduce the historical and future trends as well as interannual variabilities in precipitation, is better than with other methods that more simply regression local changes against global temperature variation (e.g. IMOGEN and OSCAR; Zelazowski et al., 2018; Gasser et al., 2017). However, the remaining biases in emulating the other three SSPs with the emulator constructed (i.e. fitted) under SSP5-8.5 imply that the sensitivity of gridded precipitation to temperature modes may have a slight dependence on SSP. There are some studies that predict an increased variability in precipitation under a warmer world (Zhang et al., 2021; Song et al., 2018), but where such additional variability is not present in the spatial temperature modes of variations. Therefore, it could be unwise to use the emulator constructed using the temperature and precipitation during the historical period or under the low emission scenarios to project future precipitation change under a high emission scenario. Constructing the emulator separately for low or high emission scenarios could help reduce uncertainties in emulating future precipitation. When coupling PREMU with other LCMs to emulate the gridded precipitation under a new prescribed emission scenario, we suggest calibrating PREMU against the SSP whose future temperature most closely resembles the temperature in LCMs under that new scenario.

### 5.3 Other versions of PREMU

### 5.3.1 PREMU constructed by different temperature lag periods

We tested the effect of different lags of gridded temperature on changes in precipitation. Previous studies noted that "memory effects" may cause precipitation to be affected more by the climate modes in the previous months (An et al., 2020). Throughout this study, we used the three-month average temperature (month of interest plus the previous two months) to capture potential lagging effects. However, to quantify the uncertainty based on duration of the lag effect, we also tested one-month temperature and six-month average temperature to construct the emulator respectively ("emulator-1mon" and "emulator-6mon"). For the

380 historical period, we found that the emulator-1mon was unable to capture the increase of GLAP after 1950. Predicting the GLAP from emulator-6mon shows a good fitting of the trends and interannual variations of GLAP in GSWP3 similar to that from emulator based on three-month average temperature, but with a lower correlation coefficient (R=0.77, p<0.001 for three-month; R=0.73, p<0.001 for six-month; Fig. S22). While for future precipitation from ESMs, all three emulators with different temperature lag periods (one-month, three-month, six-month) can capture well the changes in GLAP and gridded

precipitation under different scenarios in the future (Fig. S23, S24). This may be due to underrepresenting of topography and aerosol effects in ESMs, which are potential sources of variations in precipitation and could be important for influencing the lag-differences. Hence, we deduced that the method is highly robust (i.e. invariant) to lag-length when emulating the future precipitation (Fig. 5 and Fig. S23-24).

### 5.3.2 PREMU constructed by Tair over land

We constructed our emulator using thermal modes of variation based on the temperature from over both land and ocean grid cells. As a sensitivity study, we also evaluated the performance of the emulator constructed only using Tair over land ("emulator-land") to test its capability at predicting future precipitation. We found that the emulator-land can also reproduce consistent trends and interannual variations of GLAP and changes in gridded precipitation with ESMs (Fig. S25 and S26), while the correlation in interannual variations of GLAP with ESMs is relatively lower than PREMU ((R = 0.71 for SSP1-2.6;

R = 0.88 for SSP2-4.5; R = 0.91 for SSP3-7.0; R = 0.96 for SSP5-8.5). Considering that the change of air temperature over ocean could contain additional information that relates to local climate variability (Trenberth and Shea, 2005), we suggest retaining our inclusion of all land and ocean grid cells in PREMU calibration.

### 5.4 Potential further developments of PREMU

Our emulator has focused on total precipitation at each location. Future analyses could include testing its performance for

individual features or subsets of precipitation patterns, such as convective precipitation, large-scale precipitation and topographic precipitation. We also suggest possibly extending our emulator drivers beyond the modes of variability of air temperature only. For example, considering that the interannual variations of precipitation are mainly caused by large-scale precipitation dominated by ESNO (Cai et al., 2001; Oldenborgh and Burgers, 2005; Gupta and Jain., 2021; Zhou et al., 2020), we could consider additionally entraining sea surface temperature as a forcing variable. Similarly, for convective precipitation,

local temperature and energy for uplift could be used for prediction (Berg et al., 2013). Furthermore, PREMU may not have good capability to emulate the seasonal cycle of gridded precipitation. We suggest that a future improvement, which will allow projections at sub-yearly timescales, might be to add a residual variability module similar to that in the MESMER-M model via lag-1 autocorrelations or local spatially correlated processes (Nath et al., 2022).

**6 Conclusions**

In this study, we proposed an algorithm to construct a precipitation emulator, which could estimate gridded precipitation and its changes in a convenient and computationally effective way. We exploit strong discovered linkages between rainfall patterns and natural spatial modes of variability in near-surface temperature. To the best of our knowledge, this is a pioneer emulator that can be directly coupled within existing LCMs, and especially noting that LCMs may perform well for other variables but are currently weaker at estimating features of rainfall. This new combination will better predict global and gridded precipitation

under different emission scenarios. With illustrative examples, we demonstrated the good performance of our emulator in predicting the absolute value and interannual variations in historical precipitation from GSWP3 as well as predicted future precipitation under four scenarios from ESMs. In addition, we also verified the reliability of our results despite the potential uncertainties in the method (e.g., the assumptions of the stability in coefficient matrix in PCA and sensitivity of gridded precipitation to temperature modes). The accurate projection of future precipitation can help analyze not only expected direct

climate change under different emission scenarios but also the responses of the hydrological cycle and ecological processes to such future changes. ESMs remain hugely computationally demanding and can be only operated for a limited number of century-timescale projections. Hence robust emulators of full-complexity Earth system models remain an important tool, extrapolating ESM projections to alternative future emissions or GHG concentration scenarios that require investigation and understanding.

**Appendix 1**

| Acronym | Meaning |
| --- | --- |
| CMIP6 | The 6th phase of the Coupled Model Intercomparison Project |
| EBM | Energy Balance Model |
| ESM | Earth System Model |
| GLAP | Global land average precipitation |
| GSWP3 | Global Soil Wetness Project Phase 3 |
| IMOGEN | Integrated Model Of Global Effects of climatic aNomalies (Zelazowski et al., 2018) |
| LCM | Lower complexity models |
| MAP | Mean annual precipitation |
| MESMER | Modular Earth System Model Emulator with spatially Resolved output (Beusch et al., 2022) |
| OSCAR | A compact Earth system model (See Gasser et al., 2017) |
| PCA | Principal Component Analysis |
| PREMU | A computationally efficient precipitation emulator (This study). |
| SSP | Shared Socioeconomic Pathway |
| Tair | Surface air temperature |

**Table A1: Table of acronyms.**

## Code availability

The MATLAB code used to emulate the precipitation by PREMU is publicly available on GitHub (https://github.com/GangLiulg/PreMU), and the code used here is achieved and available on Zenodo repository (https://doi.org/ 10.5281/zenodo.7545350).

## Data availability

The GSWP3 data were available at http://search.diasjp.net/en/dataset/GSWP3_EXP1_Forcing. All CMIP6 data can be accessed from the CMIP6 archive (https://esgf-node.llnl.gov/search/cmip6/).

## Supplement

The supplement related to this article is available online at:

## Author contributions

SP conceived and designed this study. The PREMU model was coded and developed mainly by GL. GL, SP and YX prepared the paper with contributions from CH.

## Competing interests

The contact author has declared that neither they nor their co-authors have any competing interests.

## Acknowledgements

The authors would like to thank Data Integration and Analysis System (DIAS), Japan manage publicly availability of GSWP3, and CMIP6 data producers and providers.

## Financial support

This development has been supported by the National Natural Science Foundation of China (grant numbers 41722101 and 41830643).

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

| Model | Modeling Center | Variant ID |
|---|---|---|
| UKESM1-0-LL | Met Office Hadely Centre and National Environmental Research Council, UK | r1i1p1f2 |
| MPI-ESM1-2-LR | Max Planck Institute for Meteorology, Germany | r1i1p1f1 |
| MIROC6 | Japan Agency for Marine-Earth Science and Technology, Japan | r1i1p1f1 |
| IPSL-CM6A-LR | Institut Pierre Simon Laplace, France | r1i1p1f1 |
| GFDL-ESM4 | Geophysical Fluid Dynamics Laboratory, USA | r1i1p1f1 |
| EC-Earth3 | EC-Earth consortium, Europe | r1i1p1f1 |
| CanESM5 | Canadian Centre for Climate Modeling and Analysis Environment and Climate Change, Canada. | r11i1p1f1 |
| CESM2 | National Center for Atmospheric Research, Climate and Global Dynamics Laboratory, USA | r1i1p1f1 |
| ACCESS-ESM1-5 | Commonwealth Scientific and Industrial Research Organisation, Australia | r10i1p1f1 |

**Table 1: List of the 9 employed CMIP6 models and the modeling groups providing them.**

| Indice Label or Variable Symbol | Variable | Notes |
|---|---|---|
| $e$ | ESM index | Indexes nine ESMs used (see Table 1) |
| $i$ | PCA index | $i = 1$ is dominant spatial PCA, $1 \leq i \leq N_{PCA} = 10$ used |
| $k$ | Spatial index | $1 \leq k \leq N_{grid}$, where $N_{grid} = 10,368$ grids |
| $m$ | Month index | All calculations for each month, $1 \leq m \leq 12$ |
| $y$ | Year index | Index of years in periods of calibration or prediction |
| $T_{e,m}^{Timeseries}(k,y)$ | Timeseries part of PCAs, for each location, and for temperature | Average gridded Tair of three months (month of interest plus the previous two months). Note, different timeseries calculated for each location indexed by $k$. |
| $U_{e,m}^{Spatial}(k,i)$ | Spatial part of PCAs for temperature | A matrix of eigenvectors of the covariance matrix. |
| $T_{e,m}^{PCA}(y,i)$ | Temporal principal components in PCAs | Timeseries representing the characteristics of global gridded Tair |
| $P_{e,m}^{global}(y)$ | Monthly global land average precipitation for calibration | |
| $P_{e,m}^{grid}(k,y)$ | Monthly gridded (i.e. local) precipitation for calibration | |

**Table 2: List of indices labels and variable symbols in the calibration.**

| Variable Symbol | Variable | Notes |
|---|---|---|
| $T_{e,m}^{\widehat{PCA}}(y,\iota)$ | Principal components estimated by the Tair from validation datasets | Based on the assumption of constant spatial part of PCAs, $U_{e,m}^{Spatial}(k,i)$. |
| $P_{e,m}^{\widehat{global}}(y)$ | Estimation of monthly global precipitation, GLAP | |
| $P_{e,m}^{\widehat{grid}}(k,y)$ | Estimation of monthly gridded (i.e. spatial) precipitation | |

**Table 3: List of variable symbols in the validation.**

| Trend in GLAP Unit: mm yr$^{-2}$ | SSP1-2.6 | | SSP2-4.5 | | SSP3-7.0 | | SSP5-8.5 | |
|---|---|---|---|---|---|---|---|---|
| | ESM | PREMU | ESM | PREMU | ESM | PREMU | ESM | PREMU |
| ESMs average | 0.29 | 0.16 | 0.58 | 0.45 | 0.69 | 0.76 | 0.96 | 0.96 |
| UKESM1-0-LL | 0.57 | 0.30 | 0.67 | 0.53 | 0.88 | 0.87 | 1.16 | 1.16 |
| MPI-ESM1-2-LR | 0.19 | 0.06 | 0.23 | 0.26 | 0.63 | 0.69 | 0.85 | 0.85 |
| MIROC6 | 0.08 | 0.09 | 0.54 | 0.41 | 0.62 | 0.74 | 1.09 | 1.09 |
| IPSL-CM6A-LR | 0.48 | 0.27 | 0.73 | 0.56 | 1.05 | 0.90 | 1.25 | 1.25 |
| GFDL-ESM4 | 0.15 | -0.06 | 0.41 | 0.06 | -0.13 | 0.03 | 0.04 | 0.03 |
| EC-Earth3 | 0.13 | 0.18 | 0.63 | 0.72 | 1.14 | 1.21 | 1.67 | 1.69 |
| CanESM5 | 0.51 | 0.40 | 1.15 | 0.90 | 1.33 | 1.32 | 1.48 | 1.48 |
| CESM2 | 0.33 | 0.12 | 0.69 | 0.47 | 0.41 | 0.81 | 0.82 | 0.80 |
| ACCESS-ESM1-5 | 0.18 | 0.08 | 0.21 | 0.18 | 0.28 | 0.25 | 0.28 | 0.27 |

**Table 4: Trend of GLAP from ESMs average and each ESM and its corresponding emulation by PREMU during the period 2015-2100 under four scenarios. The ESMs Average represents the multi-model mean precipitation predicted by gridded temperature from 9 ESMs. Note that the unit is mm year$^{-2}$.**

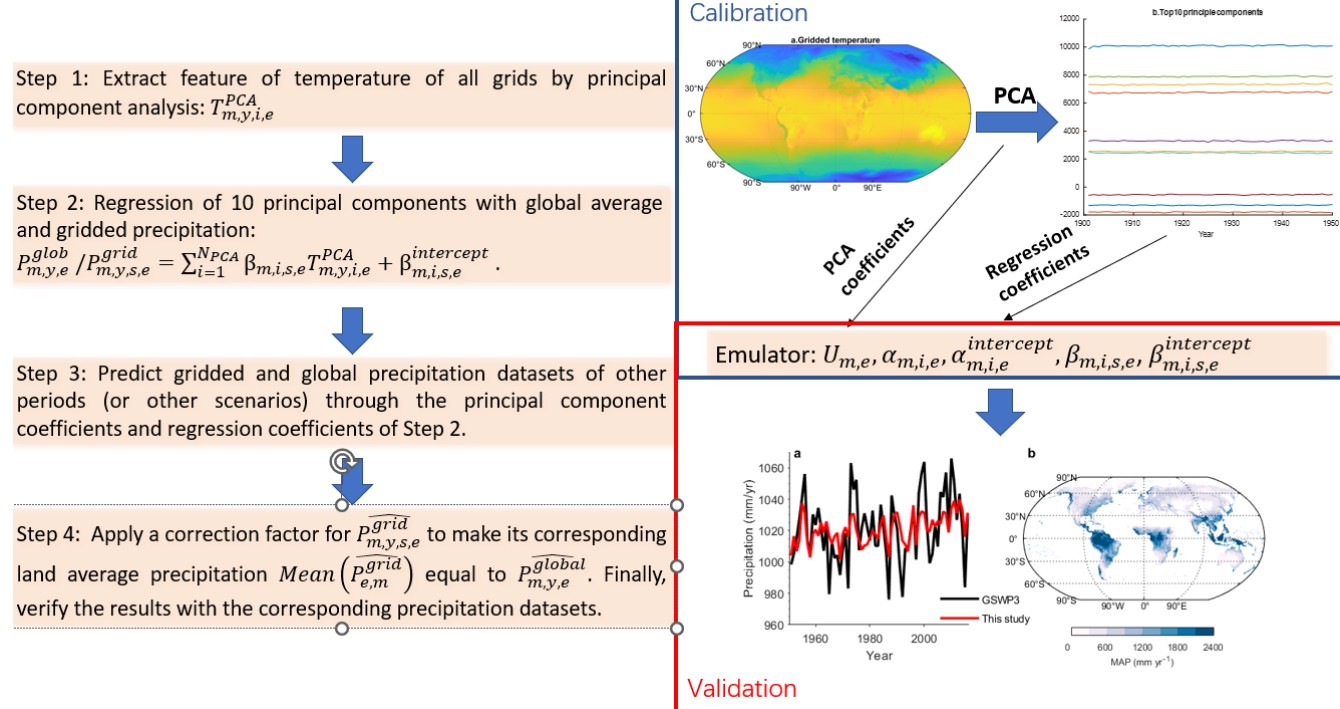

Step 1: Extract feature of temperature of all grids by principal component analysis: $T_{m,y,i,e}^{PCA}$

Step 2: Regression of 10 principal components with global average and gridded precipitation:
$P_{m,y,e}^{glob} / P_{m,y,s,e}^{grid} = \sum_{i=1}^{N_{PCA}} \beta_{m,i,s,e} T_{m,y,i,e}^{PCA} + \beta_{m,i,s,e}^{intercept}$ .

Step 3: Predict gridded and global precipitation datasets of other periods (or other scenarios) through the principal component coefficients and regression coefficients of Step 2.

Step 4: Apply a correction factor for $P_{m,y,s,e}^{\widetilde{grid}}$ to make its corresponding land average precipitation $Mean\left(P_{e,m}^{\widetilde{grid}}\right)$ equal to $P_{m,y,e}^{\widetilde{global}}$. Finally, verify the results with the corresponding precipitation datasets.

Calibration

PCA

PCA coefficients

Regression coefficients

Emulator: $U_{m,e}, \alpha_{m,i,e}, \alpha_{m,i,e}^{intercept}, \beta_{m,i,s,e}, \beta_{m,i,s,e}^{intercept}$

Validation

**Figure 1: Illustration of the precipitation emulator driven by the gridded temperature. Step 1 is to extract the features of gridded temperature through principal component analysis. In step 2, we construct the parameters of the emulator by regression of GLAP/gridded precipitation and principal components of gridded temperature. Then we use the emulator to predict the precipitation for validation period/ scenario in Step 3. Finally, we calibrate the precipitation and verify the results with precipitation from GSWP3 and ESMs. Here, $T_{e,m}^{PCA}(y,i)$ is the principal components extracted by PCA. $P_{e,m}^{global}(y)$ is the GLAP and $P_{e,m}^{grid}(k,y)$ is the precipitation a given grid $k$. $\alpha_{e,m}(i)/\beta_{e,m}(k,i)$ represents the coefficients of each principal components of temperature to global/gridded precipitation and $\alpha_{e,m}^{intercept}(i)$ /$\beta_{e,m}^{intercept}(k,i)$ is the intercept term. And $U_{e,m}^{Spatial}(k,i)$ is the PCA coefficients. Note that, the gridded temperature from GSWP3 is area-weighted averaged to 2.5° x 2.5° before PCA.**

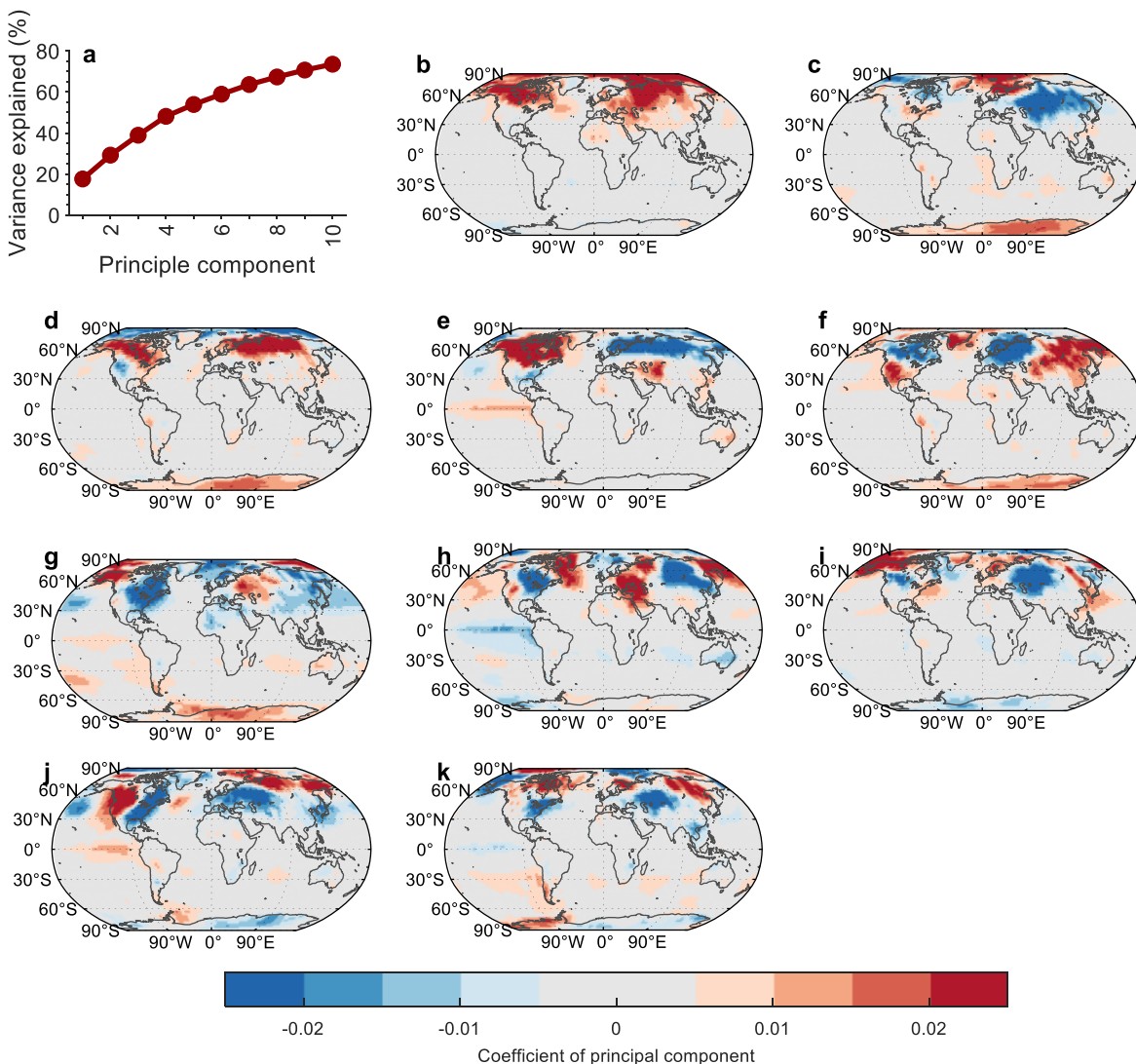

**Figure 2: The corresponding results from the PCA of the gridded temperature in January between 1901 and 1950 from GSWP3: a) The cumulative variance explanation rate of the top ten principal components. b-k) Coefficients of the top ten principal components (corresponding to $U_{e,m}^{Spatial}(:,i), i = 1 - 10$).**

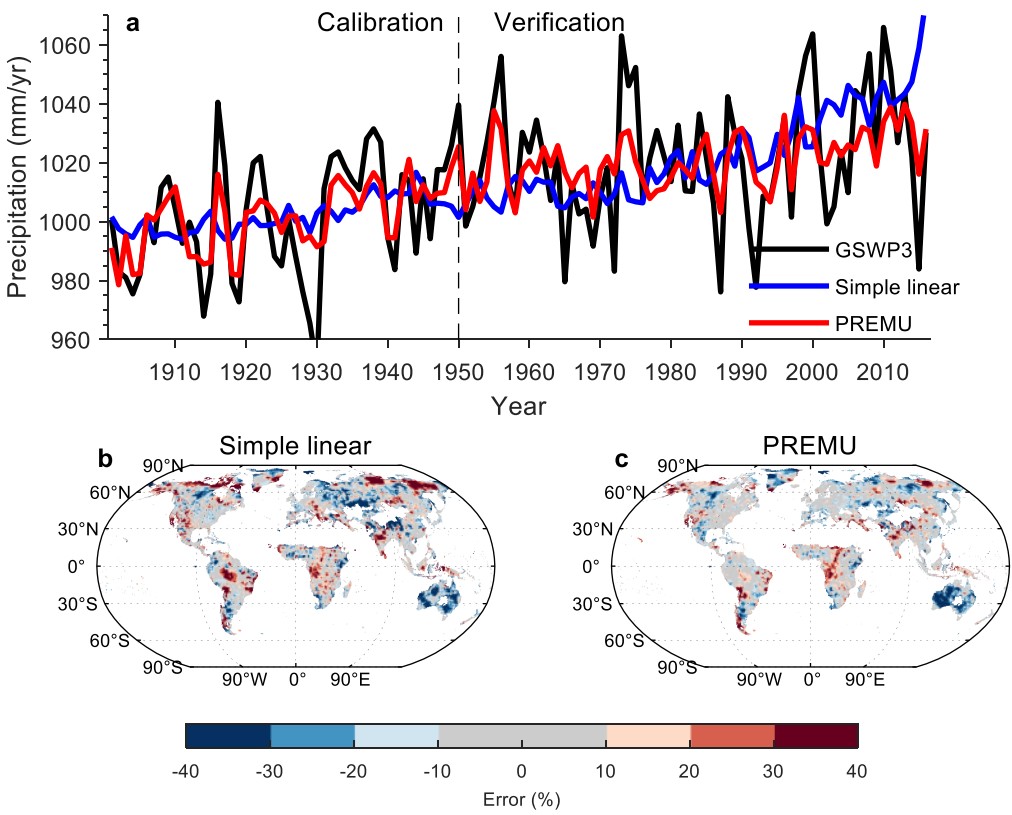

**Figure 3: The emulation on historical precipitation: a) GLAP of the historical observation precipitation (GSWP3), the prediction precipitation estimated by the simple linear method (Simple linear) and our emulator (PREMU) in 1901–2016. b) The spatial pattern of error of MAP in 1987-2016 between Simple linear and GSWP3. c) The spatial pattern of error of the MAP in 1987-2016 between our emulator and GSWP3. The global land average does not include Antarctica because of no emulation for Antarctica and the arid areas where the MAP from GSWP3 is less than 200 mm year$^{-1}$ in 1980-2016 are masked.**

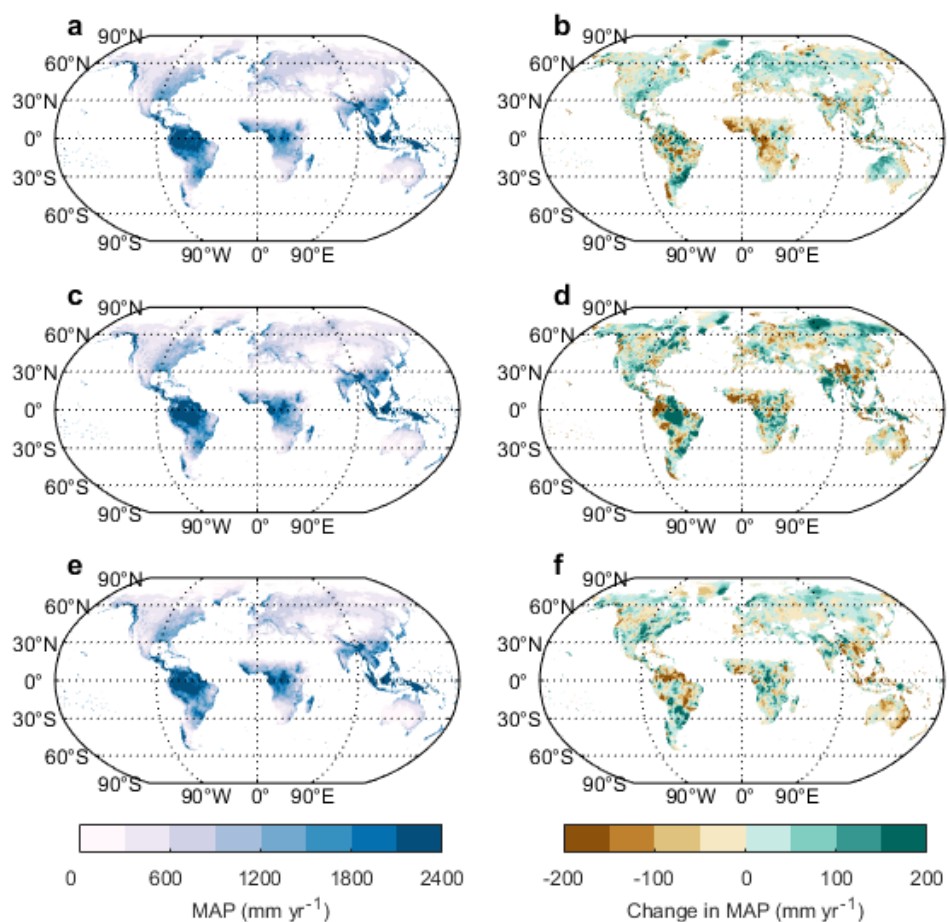

**Figure 4: Spatial pattern of MAP and change in MAP: a) The MAP in 1987-2016 from GSWP3. b) The spatial pattern of change in MAP between the period of 1951-1980 and 1987-2016 from GSWP3. c-d). Same as a-b) but for the emulation from Simple linear. e-f). Same as a-b) but for the emulation from PREMU.**

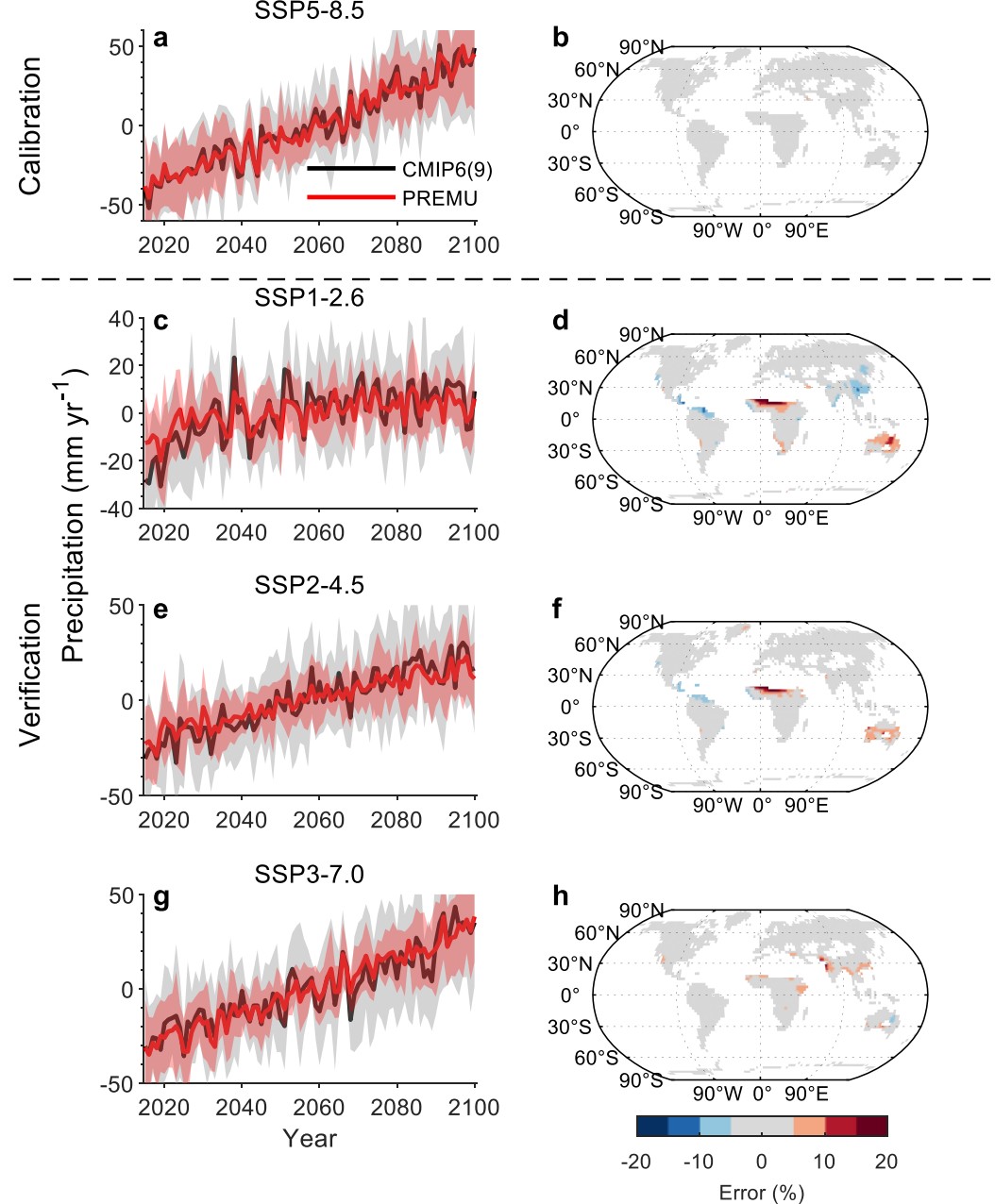

**Figure 5: The emulation on future precipitation: a) multi-model mean GLAP in 9 ESMs from CMIP6 (CMIP (9)), and the**
665 **precipitation prediction by our emulator (PREMU) in 2015-2100 under the SSP5-8.5 scenario. The shaded area represents the**
**mean±std. b) The spatial pattern of error in MAP during 2071-2100 between multi-model mean and our emulator. c-d) SSP1-2.6; e-**
**f) SSP2-4.5; g-h) SSP3-7.0.**

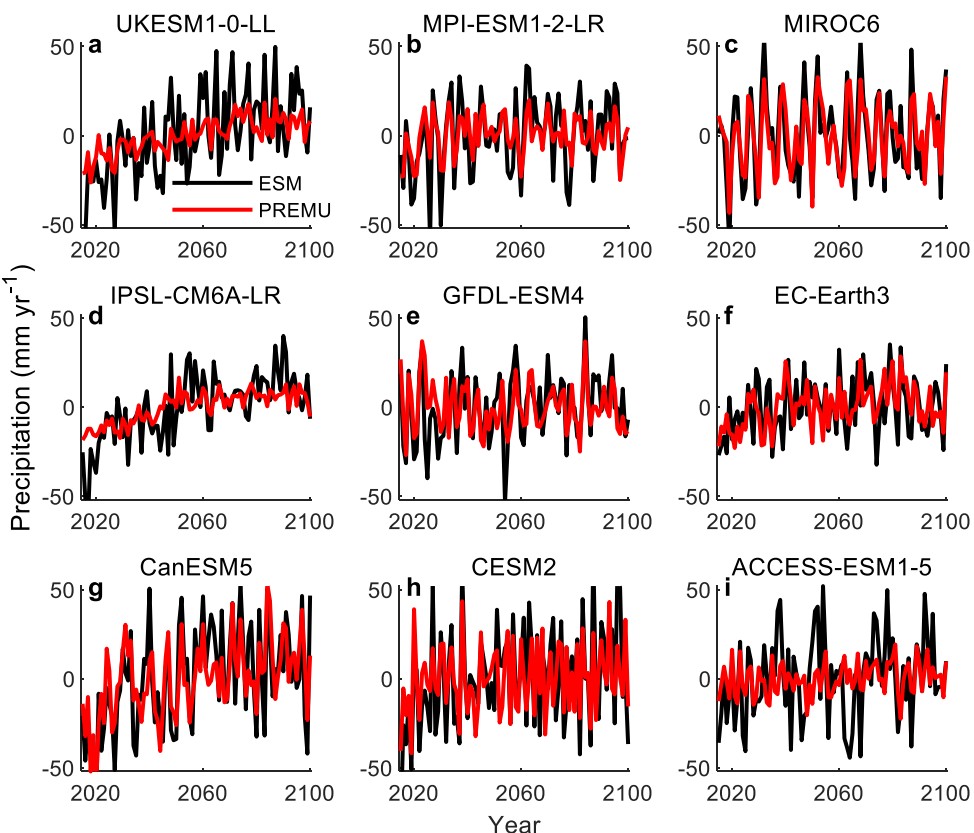

**Figure 6: The anomaly of GLAP from each ESM under the SSP1-2.6 scenario: a) UKESM1-0-LL; b) MPI-ESM1-2-LR; c) MIROC6; d) IPSL-CM6A-LR; e) GFDL-ESM4; f) EC-Earth3; g) CanESM5; h) CESM2; i) ACCESS-ESM1-5.**

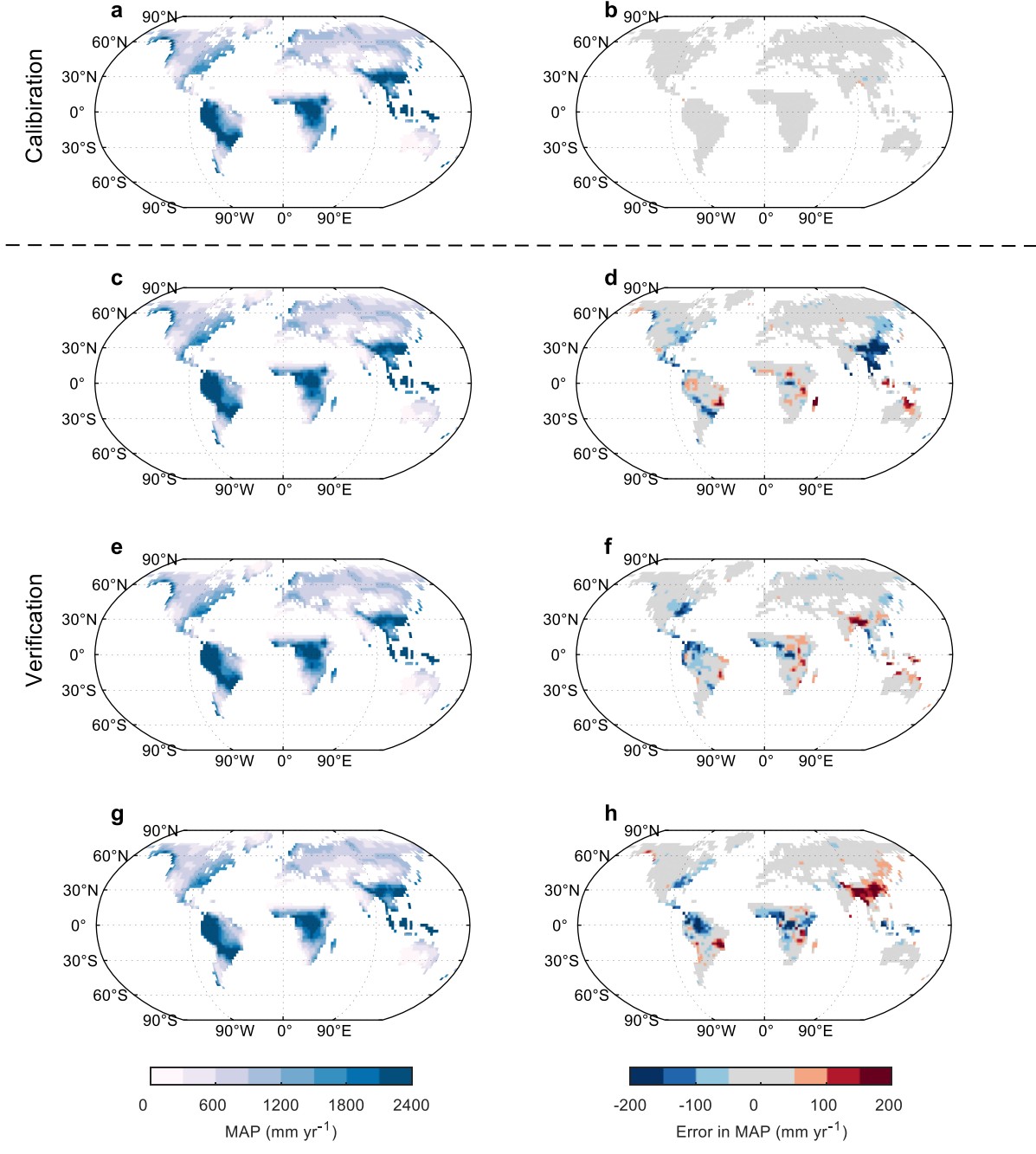

Figure 7: The spatial pattern of MAP in 2071-2100 from a) multi-model mean of ESMs and b) the error in MAP from emulation from PREMU. c-d) SSP1-2.6; e-f) SSP2-4.5; g-h) SSP3-7.0.

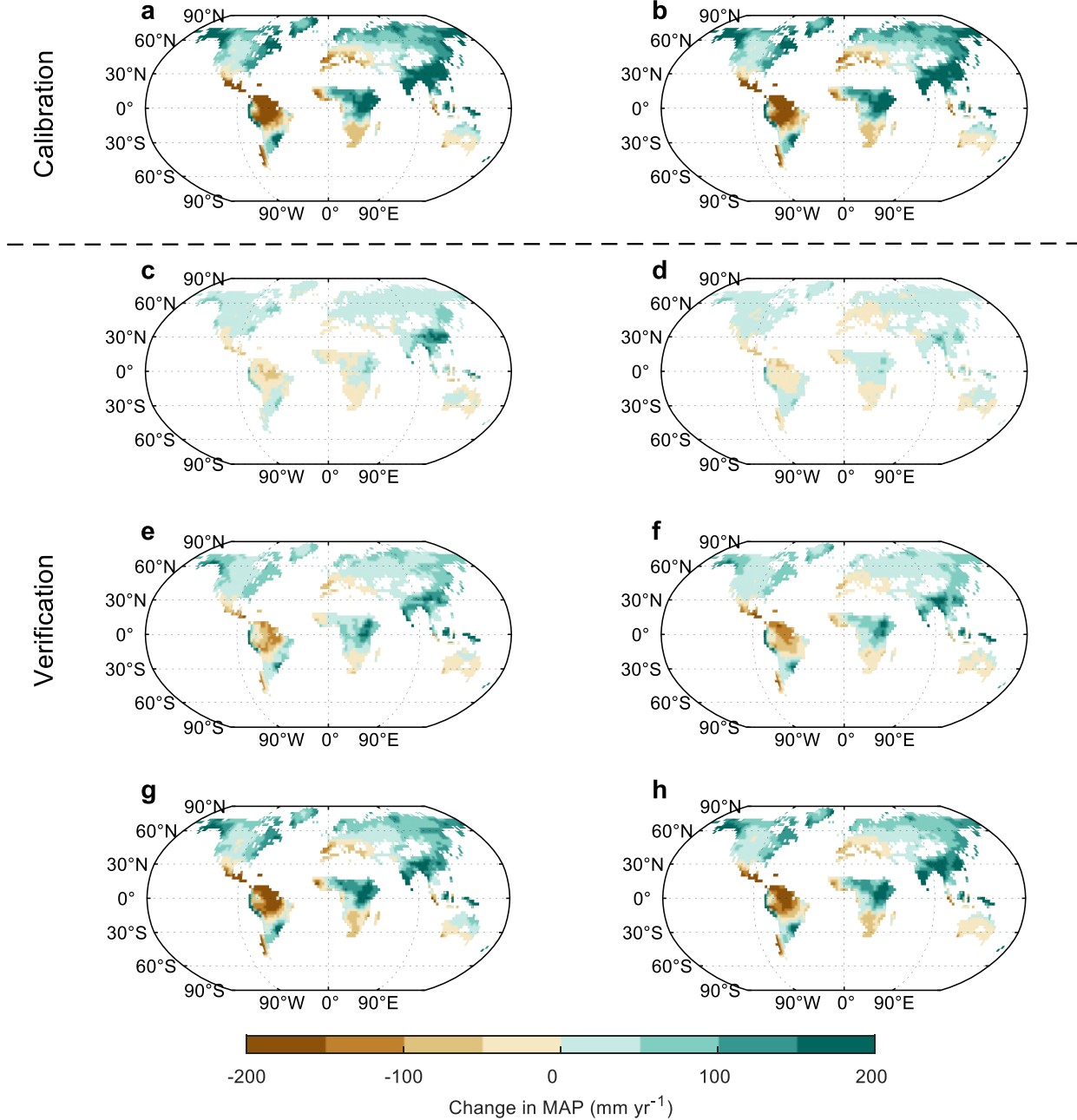

**Figure 8: The spatial pattern of change in MAP in 2071-2100 from a) multi-model mean of ESMs and b) emulation from PREMU**
**under SSP5-8.5 scenario. c-d) SSP1-2.6; e-f) SSP2-4.5; g-h) SSP3-7.0.**

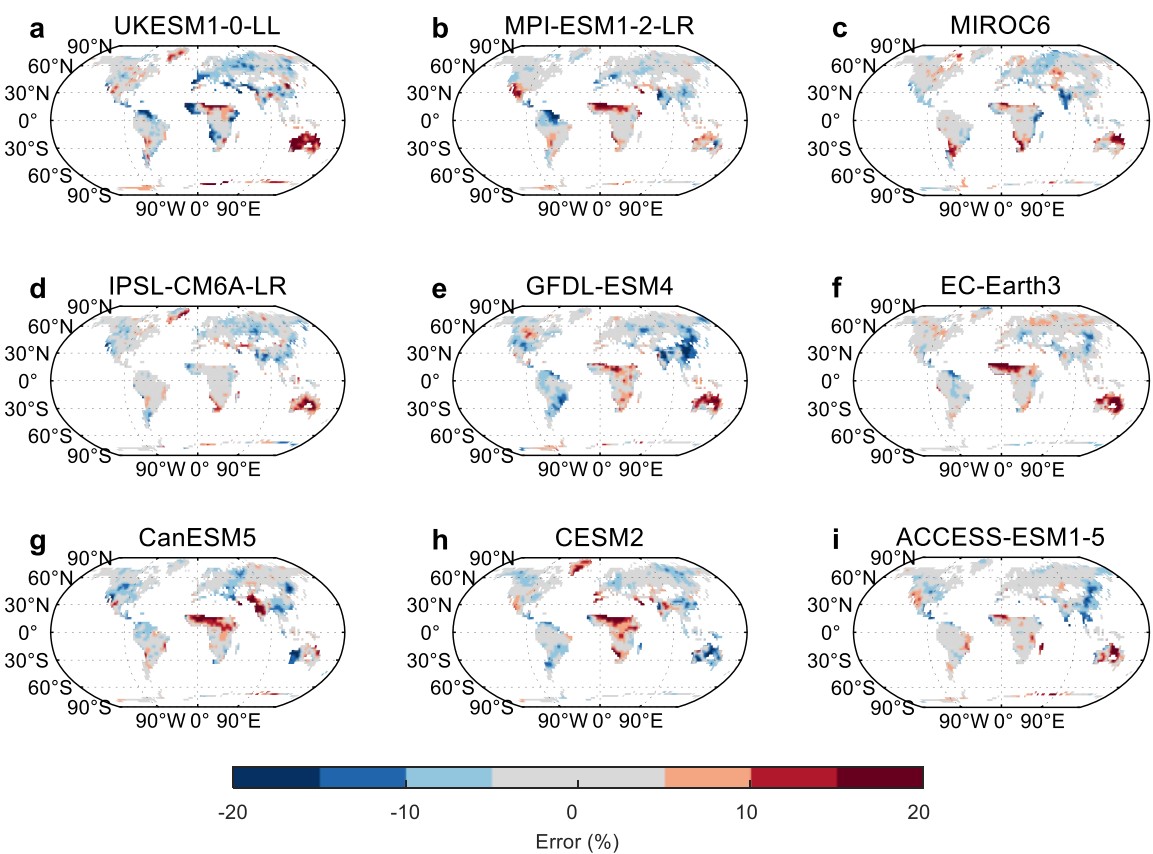

**Figure 9:** **The spatial pattern of error of the MAP in 2071-2100 between each ESM and emulation from PREMU under the SSP1-2.6 scenario: a) UKESM1-0-LL; b) MPI-ESM1-2-LR; c) MIROC6; d) IPSL-CM6A-LR; e) GFDL-ESM4; f) EC-Earth3; g) CanESM5; h) CESM2; i) ACCESS-ESM1-5.**

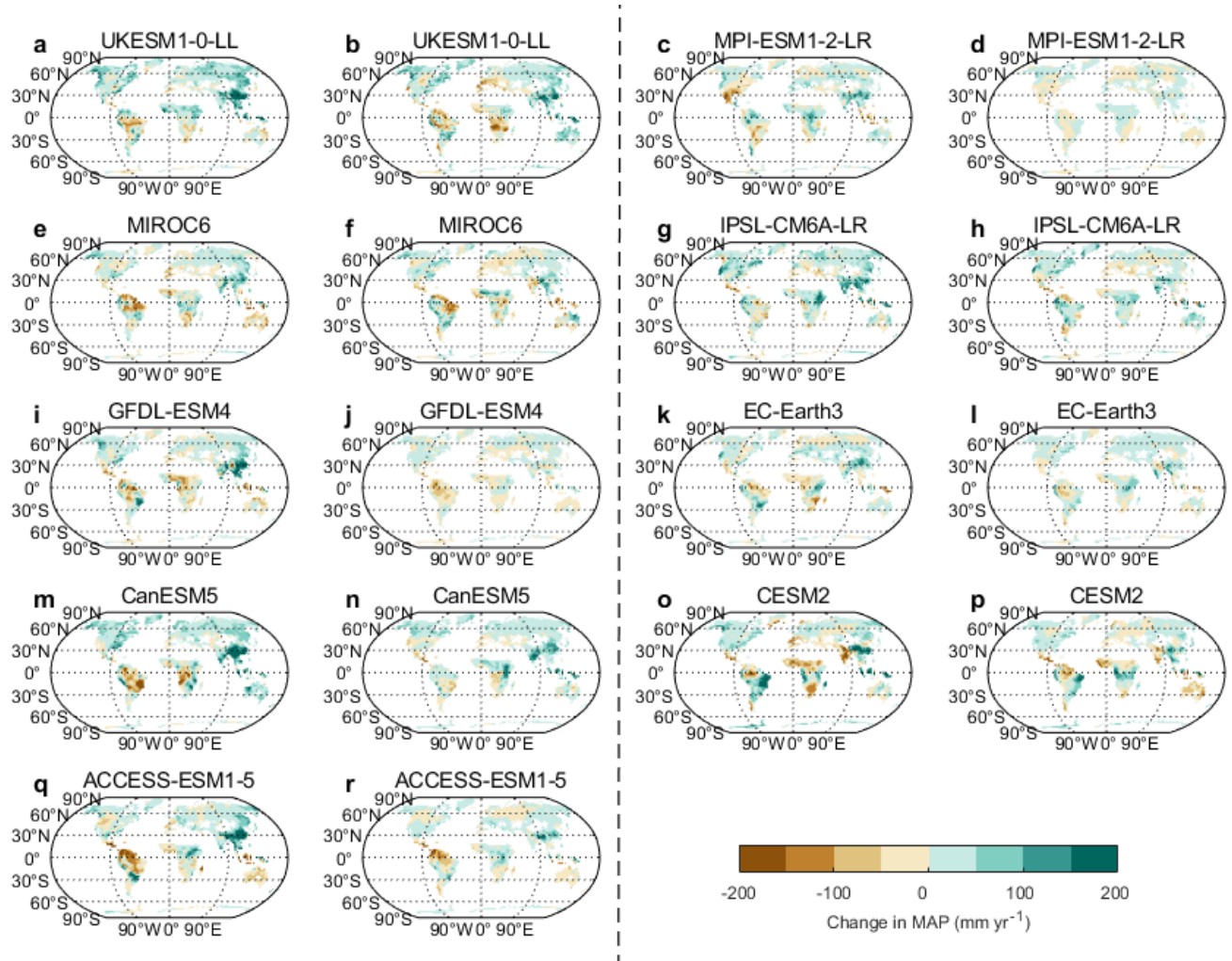

**Figure 10: The spatial pattern of change in MAP between the period of 2015-2044 and 2071-2100 from a) UKESM1-0-LL and b) emulation from PREMU under SSP1-2.6 scenario. c-d) MPI-ESM1-2-LR; e-f) MIROC6; g-h) IPSL-CM6A-LR; i-j) GFDL-ESM4;**
**k-l) EC-Earth3; m-n) CanESM5; o-p) CESM2; q-r) ACCESS-ESM1-5.**

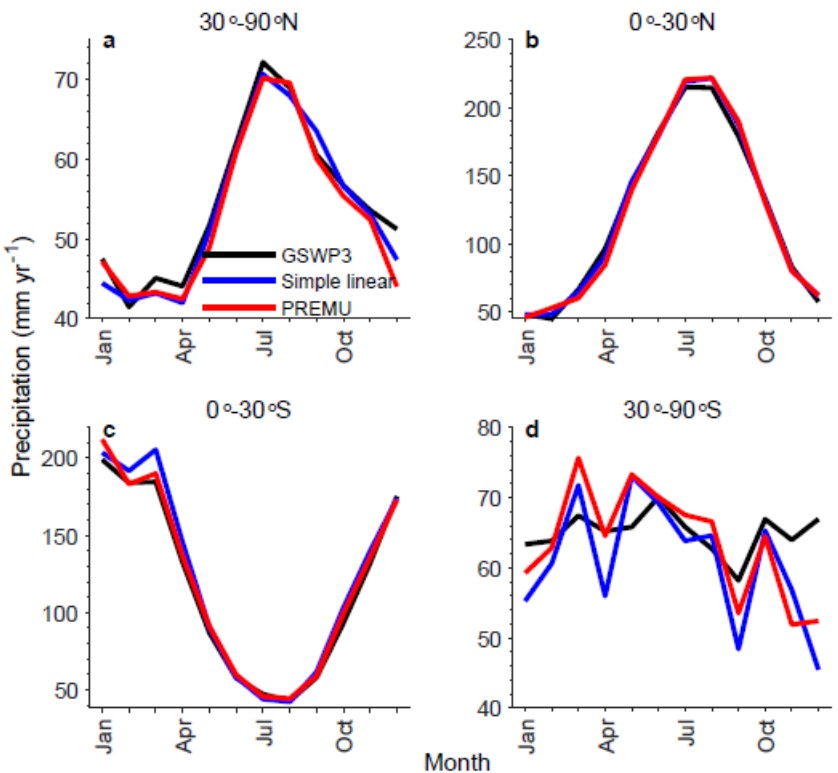

**Figure 11: a) The seasonal cycle of 30° N-90° N latitude land average precipitation in 1987-2016 from GSWP3, simple linear method and PREMU; b) Eq.-30° N; c) 30° S-Eq.; d) 90° S-30°S.**