# Peer review of "A new precipitation emulator (PREMU v1.0) for lower complexity models"

_Geoscientific Model Development, 2022_

## Author Comment (AC1)

**Author's Response to the Review Comments**

We appreciate the time and effort by the Editor and reviewers in reviewing our manuscript and the thoughtful and valuable suggestions offered. We have attempted to address in full all issues raised by reviewers, and we hope that the revised manuscript is now to their satisfaction and can be considered for publication in GMD. In the following reply, the black italic text are the reviewers requests, the black indented text are our responses and the blue text in quotes are revised text in the manuscript.

**1. Response to Comments from Reviewer #1:**

*PREMU presents a novel method for emulating precipitation by considering modes of temperature variability, thus providing a more precise approach towards representing such a dynamic climate variable. I believe this paper would be useful for the GMD reader base and provide new avenues for tackling emulation of other climate variables that do not scale so directly with Global Mean Temperature. In general, the text could undergo some restructuring for understandability and clarity purposes with some suggestions to do so provided in the specific comments below.*

> **[Response]** We thank the reviewer for the positive feedback. We are particular pleased that our method is regarded as "novel" and that it offers a more "precise approach" compared to standard pattern-scaling in global temperature. We appreciate the suggestions to help raise clarity and understandability, and we list below how we have adopted those suggestions.

*Some more general comments on the methodological approach that could also be improved upon are as follows:*

*1. The methods overall make sense but could be more explicit in terms of the PCA analysis done: how are the spatial patterns derived, how sensitive are these patterns to sample size, do these patterns make sense/have physical explanations behind them that are identified in previous literature. Additionally, there are parts of the discussion where it is mentioned that the sensitivity of the PCA based coefficient matrix is tested (L271-L281), this could be mentioned before as a subsection under Methods and Results.*

> **[Response]** Thank you for this constructive suggestion. The spatial pattern means that the spatial pattern of component coefficients $U_{e,m}^{Spatial}$. We have adjusted the paper to explain more clearly that how $U_{e,m}^{Spatial}$ is derived in the Calibration section: "The PCA component coefficients, $U_{e,m}^{Spatial}(k,i)$, are the combination of eigenvectors of the covariance matrix of the anomalies of the gridded 3-month average Tair.". Then we used the "leave-one-year out" method to evaluate the sensitivity of $U_{e,m}^{Spatial}$ to samples. Taking the coefficients in July as an example (Fig. R1), with a "leave-one-year out" method, we find that the standard deviations of coefficients are quite small, which confirms the robustness of the PCA method. For the physical explanations of the coefficients, we agree it is interesting and

worthy of mentioning: "Previous studies have confirmed the important impact of ocean-atmosphere oscillations on inter-annual and inter-decadal variations of regional precipitation (e.g., Li et al., 2021; Tsanis and Tapoglou, 2019; Dai, 2013). As such, our noted changes in gridded surface air temperature likely additionally contain information about sea surface temperature and ocean-atmosphere oscillations, as additional information on the background global land average temperature.". We hypothesize that these PCs may be linked to ocean-atmosphere oscillations. For the inter-annual modes of variability, we found that each climate index has a correlation coefficient >0.4 or <-0.4 with at least 2 PCs (Fig. R2, also shown as Fig. S1 now). Using the ten PCs as independent variables, the 15 climate indices can be well reconstructed by multi-linear regression ($R^2$ = 0.40 ~ 0.94). This suggests the ten PCs could substantially represent climate ocean-atmosphere oscillations, and we now writing: "Otherwise, we found that the 15 ocean-atmosphere climate indices can be well reconstructed by multi-linear regression of the top 10 principal components $T_{e,m}^{PCA}(y, 1:10)$ (Fig. S1). This suggests that the principal components of gridded temperature likely additionally contain information about sea surface temperature and ocean-atmosphere oscillations, which have great impacts on inter-annual and inter-decadal variations of regional precipitation.". The physical explanations of these coefficients need further study in future: "Furthermore, we have assumed that the coefficients of PCA are linked with the ocean-atmosphere oscillations, but the detailed physical explanations of the coefficient matrix need further study in the future.". For the sensitivity of $U_{e,m}^{Spatial}$ to different scenarios, we have evaluated it detailed in Discussion, but alert in Methods.

[Figure]

Standard deviations of 50 coefficients from leave-one-out approach

**Figure R1** The standard deviations of the 50 coefficients of the top ten PCs in July derived from the "leave-one-year out" approach.

[Figure]

**Figure R2** A heat map of the correlation coefficients between the annual mean of 10 PCs and 15 location-based climate indices from 1950 to 2016. The 15 climate indices are the Pacific North America Index (PNA); East Pacific/North Pacific Oscillation (EPNP); Western Pacific Index (WP); Eastern Asia/Western Russia (EAWR); North Atlantic Oscillation (NAO); two ENSO indices (Eastern Tropical Pacific SST, NINO 3; East Central Tropical Pacific SST, NINO 3.4); Tropical Northern Atlantic Index (TNA); Tropical Southern Atlantic Index (TSA); Western Hemisphere warm pool (WHWP); Oceanic Nino Index (ONI); Pacific Decadal Oscillation (PDO); North Pacific pattern (NP); Trans-Nino Index (TNI); Global Mean Land/Ocean Temperature Index (Mean T index). Data is available at https://psl.noaa.gov/data/climateindices/list/.

*2.   The 3-month mean of temperature does not necessarily mean that lag effects are captured, and the authors should elaborate on this choice as compared to e.g. a multi-linear regression. Again the discussion brings about some tests of lag effects using premu-1mon and premu-6mon but this still considers monthly averages, furthermore this part would also fit better as a subsection under methods and results. In terms of assessing memory effects, the evaluation could be enriched by, for instance, looking at lag-1,2 and 3 correlation coefficients between subsequent month values unless the authors are not interested in preserving month-to-month correlations within PREMU predictions which should then be justified.*

    **[Response]** Yes, we agree that the 3-month mean of Tair does not imply there are lag effects. We did consider the regression approach. However, because the regression is only based on 50 samples (or 86 samples for ESMs), if we applied a multilinear regression for

the 30 PCs derived from the month plus the previous two months, the regression will only have 20 degrees-of-freedom, which is too small given the 50 datapoints. Moreover, collinearity of the 30 PCs of the three months can also be a serious problem in the multilinear regression (Fig. R3). However, we tried the following two ways to test the lag effects of Tair on GLAP. First, we used Tair of month of interest (PREMU-1mon) and previous two months (Lag$_1$ month and Lag$_2$ month) to construct PREMU respectively (Fig. R4). In essence, the red curve of Figure R4 is in general closer to the black data-led curve, compared to the other options. During calibration period of GSWP3 (1901-1950), the correlation coefficients between GWSP3 precipitation and the reconstructed precipitation by PREMU-1mon (r=0.75) is higher than that reconstructed by the lag months (r=0.70 for Lag$_1$ month; r= 0.69 for Lag$_2$ month). However, all the three methods (PREMU-1mon, Lag$_1$ month and Lag$_2$ month) are unable to capture the increase trend of GLAP in the validation period, when compared to PREMU-3mon (Fig. R4). As well as illustrating the lag effects of PCs (Fig. R3), we noticed additionally that the lag effects decrease with month, as $R^2$ for most PCs in Lag$_2$ month is lower than that of Lag$_1$ month (Fig. R3).

As a second test of lag effects, we extracted the 10 PCs by applying PCA for the month plus the previous two months (named PREMU-newPCs):

$$T_{e,m}^{PCA}(y,i) = \sum_{k=1}^{N_{grid}} U_{e,m}^{Spatial}(k,i) * T_{e,m}^{Timeseries}(k,y) + \sum_{k=1}^{N_{grid}} U_{e,m-1}^{Spatial}(k,i) * T_{e,m-1}^{Timeseries}(k,y) +$$

$$\sum_{k=1}^{N_{grid}} U_{e,m-2}^{Spatial}(k,i) * T_{e,m-2}^{Timeseries}(k,y)$$

Similarly, this method gives a correlation coefficient of only 0.69 between GWSP3 and reconstructed precipitation during the calibration period, and cannot capture the increasing trend of GLAP after year 1950, as noted in the GSWP3 database (Fig. R5). We noticed that this may due to the lower ratio of cumulative variance explanation rate of the top 10 PCs 54% - 59%, compared to 65%-74% in the PREMU based on 3-month mean Tair. The performance of PREMU-3mon is better than the emulator based on Tair of just one month (either PREMU-1mon, or Lag$_1$ month or Lag$_2$ month) and the emulator constructed by extracting the 10 PCs from Tair of 3 months (PREMU-newPCs). Thus, we suggest it is valid to use PREMU-3mon to emulate the precipitation. After careful consideration, we decided to leave this paragraph in Discussion as a subsection. We have telegraphed in the Method that the choice of 3-month average Tair will be illustrated in Discussion "The choice of the 3-month average Tair as independent variables is robust, with further details illustrated in Sect. 5."

[Figure]

**Figure R3** The $R^2$ in reconstructing the top 10 PCs in Lag$_1$ month and Lag$_2$ month by the multi-linear regression of 10 PCs in PREMU-1mon.

[Figure]

**Figure R4** Same as Fig. 3 but for the PREMU based on Tair of month of interest (PREMU-

1mon) and previous two months (Lag$_1$ month and Lag$_2$ month).

[Figure]

**Figure R5** Same as Fig. 3a but for the PREMU based on 10 PCs derived from 3 months Tair (PREMU-newPCs) rather than 3-month mean Tair (PREMU-3mon).

*3.    The results only focus on MAP, inter-annual variations and trends and annual spatial patterns, given that PREMU is monthly, readers would also benefit from seeing these analyses done on a seasonal/monthly level which would also be more impact relevant.*

**[Response]** Thank you for requesting, we show the performance of emulating precipitation at the seasonal/monthly timescale. PREMU could capture the seasonal cycle of GLAP in different latitude bands (Fig. R6), and spatial patterns of seasonal precipitation with little error in specific locations such as the boreal regions. However, elsewhere the errors were large e.g. with 18-22 mm mon$^{-1}$ error in tropics (Fig. R7, also shown as Fig. S16). In the revised version, we have added a subsection of seasonal cycle (Sect 4.3): "As a monthly emulator, the performance of PREMU at describing the seasonal cycle of precipitation requires evaluation. For the historical observations, PREMU can capture the seasonal cycle of GLAP in each latitude band (Fig. 11). There are some spatial differences, with little error in boreal regions but with 18-22 mm mon$^{-1}$ error in tropics (Fig. S16).    We found fewer grid cells showing more than 20 mm mon$^{-1}$ error of seasonal precipitation from GSWP3 in our emulation (18% - 24%) compared to using a simple linear fit (24% - 31%; Fig. S16). As for the future precipitation, PREMU shows a good performance on emulating the monthly land average precipitation in each latitude band under all scenarios (Fig. S17). However, PERMU tends to overestimate the JJA (June, July, August) and SON (September, October, November) precipitation in South Asia and Amazon and underestimates the SON and DJF (December, January, February) precipitation in West Africa under SSP1-2.6, while the error in spatial pattern of seasonal precipitation decreases from low to high emission scenarios (Fig. S18)."*. In MESMER-M (an emulator for spatially resolved monthly temperature; Nath et al., 2022), residual variability module can improve its ability to capture the monthly variations of temperature. For precipitation, adding the residual variability module may further improve the performance on seasonality. Thus, we have added discussion in Sect. 5: "Furthermore, PREMU may not have good capability to emulate the seasonal cycle of gridded precipitation. We suggest that a future improvement, which will allow projections*

at sub-yearly timescales, might be to add a residual variability module similar to that in the MESMER-M model via lag-1 autocorrelations or local spatially correlated processes (Nath et al., 2022)."

[Figure]

**Figure R6 a)** The seasonal cycle of 30° N-90° N latitude land average precipitation in 1987-2016 from GSWP3, simple linear method and PREMU; **b)** Eq.-30° N; **c)** 30° S-Eq.; **d)** 90° S-30° S.

[Figure]

**Figure R7 a)** The spatial pattern of error of the three months average (March, April and May) in 1987-2016 between simple linear method and GSWP3; **b)** The spatial pattern of error of the spring average precipitation in 1987-2016 between PREMU and GSWP3; **c-d)** JJA (June, July, August); **e-f)** SON (September, October, November) ; **g-h)** DJF (December, January, February).

**Specific Comments:**

*(1) L8: Elaborate on the term rainfall features, otherwise it is a bit vague.*
    **[Response]** We now write more explicitly as: "Rainfall features (e.g., strength or frequency)".

*(2) L20: MESMER-M is a monthly extension of MESMER, given that PREMU takes monthly input this may be relevant to mention (Nath et al. https://doi.org/10.5194/esd-13-851-2022)*

**[Response]** Thanks. This reference has been added.

*(3) L23-26: consider moving the policy relevance of LCMs higher up in the abstract, the final sentences should focus on PREMU itself.*

**[Response]** Thanks. It has been moved higher up in the Abstract: "Some current lower complexity models (LCMs) are designed to emulate precipitation in a computationally effective way. However, for precipitation in particular, they are known to have large errors, due to their simpler linear scaling of precipitation changes against global warming (e.g., IMOGEN; Zelazowski et al., 2018).". In addition, we focus on PREMU in the final sentence: "By coupling with PREMU, LCMs will have the ability to emulate gridded precipitation, thus they can be widely coupled with hydrological models or land surface models.".

*(4) Introduction: Generally a good overview into the problem this study is trying to tackle, however it tends to downplay the real value of this work and could benefit from some restructuring to make the points of this study clearer:*

**[Response]** Thanks for the suggestion. We have carefully restructured the Introduction following the reviewer's suggestion.

*(4.1) L28-L32: Consider merging this with the next paragraph, otherwise it comes across as a bit redundant.*

**[Response]** Thanks. It has been merged with the next paragraph.

*(4.2) L34-L36: Paints a good picture of LCMs, in relevance to this study and generally for LCMs it should be emphasised that they focus on reproducing only a few climate variables that are most impact relevant. This allows them to have such reduced computational expenses as well as differentiates them from ESMs. If this is what is meant by "highly parameterized macro-properties" it should still be elaborated upon, in general this term can imply a lot of things so it should be made more specific.*

**[Response]** We have removed the use of words "macro-properties". We also add that LCMs can focus on reproducing only a few climate variables that have the most relevance for climate change impacts. We now write: "Thus, Lower complexity models (LCMs) are designed as the common approaches to improve computational efficiency in climate change research to focus on the most impact relevant variables".

*(4.3) L42: The following reference could be insightful for the author/readers when mentioning pattern scaling:*

*Tebaldi, C., & Arblaster, J. M. (2014). Pattern scaling: Its strengths and limitations, and an update on the latest model simulations. Climatic Change, 122(3), 459–471. https://doi.org/10.1007/s10584-013-1032-9*

*Tebaldi, C., & Knutti, R. (2018). Evaluating the accuracy of climate change pattern emulation for low warming targets. Environmental Research Letters, 13(5). https://doi.org/10.1088/1748-9326/aabef2*

**[Response]** These references have been added: "Pattern-scaling multiplies global temperature change by spatial patterns to give regional information (Zelazowski et al., 2018; Gasser et al., 2017; Huntingford et al., 2010; Tebaldi and Arblaster, 2014; Tebaldi and Knutti, 2018 ).".

*(4.4) It could be good to add that there is already some work that successfully considers modes of variability within emulation, see:*
*Mckinnon and Deser 2018: https://doi.org/10.1175/JCLI-D-17-0901.1*
*Mckinnon and Deser 2021: DOI: 10.1175/JCLI-D-21-0251.1*

**[Response]** These references have been added: "Recently, a spatially resolved emulator (MESMER) coupled with the emission-driven LCM (MAGICC) has been developed, which is found to reproduce the spatial pattern of temperature in ESMs better (Beusch et al., 2022; Mckinon and Deser 2018; Mckinon and Deser 2021).".

*(4.5) L45: I feel like the fact that precipitation is a crucial variable but has statistical features that make its representation within traditional LCM approaches difficult is a strong message to highlight. Consider restructuring the first three paragraphs to emphasise this i.e.*
*o First paragraph:*

*There is increasing demand for climate information however lack of sufficient computational power for running ESMs for all potential future emission scenarios etc. .*

*LCMs are a solution to this representing key climate variables, however in terms of gridded data most use pattern scaling based off GMT which is suited for local temperatures.*
*o Second paragraph:*

*Precipitation has high spatio-temporal variability and is affected by atmospheric dynamics, inter-annual modes of variability etc. making representation within LCMs using only GMT as an explanatory variable difficult e.g. OSCAR, IMOGEN*

*Nevertheless precipitation is a crucial component of the water cycle (Eltahir and Bras, 1996; Trenberth et al., 2003; Sun et al., 2018), has key societal implications and is closely associated with the functioning of terrestrial ecosystems*

**[Response]** Thanks. The first three paragraphs have been reconstructed following the reviewer's suggestions. We also highlighted that precipitation is a crucial variable but has statistical features that make its representation within traditional LCM approaches difficult in the second paragraph (L49- L67): "Precipitation has high spatio-temporal variability and is affected by atmospheric dynamics, inter-annual modes of variability (Li et al., 2021; Tsanis and Tapoglou, 2019), making representing within tradition al LCM approaches difficult. Thus, only two LCMs (IMOGEN and OSCAR) have tried to emulate precipitation but not well (Zelazowski et al., 2018; Gasser et al., 2017). IMOGEN emulates the gridded precipitation based on the regression relationship (by month and location) between gridded precipitation and global land average temperature (Zelazowski et al., 2018). OSCAR constructs the emulator by establishing the relationship between global average precipitation and global average temperature and radiative forcing, and then using the pattern-scaling method to deduce the gridded precipitation (Gasser et al., 2017). Nevertheless, the gridded precipitation estimated by the simple linear

method is not reliable either in IMOGEN or in OSCAR; the gridded precipitation predicted by IMOGEN only explains less than 20% of variance of seasonal precipitation in most regions (Zelazowski et al., 2018) and OSCAR cannot capture the interannual variations of regional precipitation across the globe at all (Gasser et al., 2017). This may be because 1) the global average temperature could not fully characterize local temperature and moisture recycling (Prein et al., 2017) and large-scale circulation (Shepherd, 2014; Fereday et al., 2018; Heinze-Deml et al., 2021); 2) there is at any given location a nonlinear relationship between local rainfall features and global warming (Allen and Ingram., 2002; Collins et al., 2013; Chadwick and Good, 2013). Hence models such as IMOGEN, that rely on linear pattern-scaling, by definition cannot fully capture expected future precipitation changes. Nevertheless, precipitation is a crucial component of the water cycle (Eltahir and Bras, 1996; Trenberth et al., 2003; Sun et al., 2018), has key societal implications and is closely associated with the functioning of terrestrial ecosystems. Accurately emulating precipitation change is necessary to determine the climate response to different emission scenarios and to understand feedbacks to global warming via rainfall-dependent vegetation net primary productivity (Gasser et al., 2017).".

*(4.6) L65-L67: Here it may also be a good idea to elaborate on the rainfall features this study finds especially relevant to preserve and provide relevant literature references (is it just inter-annual variability, spatial and seasonal variance, or are there more to think of under a changing climate? What are the drivers of inter-annual variability in precipitation e.g. El-Nino, Indian Ocean Dipole?) and why basing off gridded temperature provides a more promising avenue as compared to GMT to achieve this.*

**[Response]** Thanks. Previous studies suggest that a substantial amount of the inter-annual variability of regional precipitation is linked to key oceanic oscillations (Li et al., 2021; Tsanis et al., 2018; Dai et al., 2013). For example, precipitation in Europe is more connected with North Atlantic Oscillations, and US precipitation is linked with the Pacific Oscillation. In our study, the top ten PCs based on grided temperature can reconstruct most of the ocean-atmosphere index during 1950-2021 (Fig. R2, also shown as Fig. S1 now). We have revised the manuscript following the above suggestions: "Previous studies have confirmed the important impact of ocean-atmosphere oscillations on inter-annual and inter-decadal variations of regional precipitation (e.g. Li et al., 2021; Tsanis et al., 2018; Dai et al., 2013). As such, our noted changes in gridded surface air temperature likely additionally contain information about sea surface temperature and ocean-atmosphere oscillations, as additional information on the background global land average temperature."

*(4.7) L66: The end-use of the resulting emulator should already be introduced, it seems like it can have multiple uses in terms of looking at future emission pathways e.g.*
*o   Stand in for OSCAR and IMOGEN's precip. pattern scaling (here it should be noted that these LCMs have interactive atmospheric chemistry and endogenous calculations of e.g. biomass burning which may lead to differences from ESMs in the PCA analysis and therefore the resulting emulator is not an ESM one but simply a module within the aforementioned LCMs)*
*o   An extension of the MAGICC-MESMER emulation chain.*

**[Response]** Yes, we agree that this note will let the reader better understand the meaning of PREMU. We have added this in the revised version: "We have designed the presented emulator such that it can act as, for instance, an enhanced precipitation module for the OSCAR and IMOGEN models. Alternatively, PREMU could be coupled directly with the MAGICC-MESMER and MESMER-M emulators."

*(5) L70: Consider making Data and Methods two independent sections, it seems like they have that role in the text anyways.*

    **[Response]** OK.

*(6) L71: The text structure could benefit from introducing the approach before going into the subsections i.e. 1) you divide the emulator calibration into that done on the historical period 1901-2016 and that done on the future period 2016-2100 2) for the historical calibration, you use observational data while for future calibration you use ESM data.*

**[Response]** Thanks. We added the introducing of the approach before the subsections: "In the analysis, we tested the performance of PREMU for both historical periods (1901-2016) and future periods (2016-2100). For historical periods, we used different time periods of observation data for calibration (1901-1950) and validation (1951-2016). While for the future, we used ESM data under different emission scenarios for calibration (SSP5-8.5) and validation (SSP1-2.6, SSP2-4.5 and SSP3-7.0).".

*(7) L72-73: the use of "first tested" is a bit misleading since you both calibrated and tested PREMU on historical data, maybe replace with "we first demonstrate the applicability of the emulator on observational data provided by the Global Soil Wetness Project Phase 3."*

    **[Response]** It has been revised and following the proposed wording.

*(8) L77: The statement: 'yet provides additional high frequency signals which are lacking in previous products' leaves the reader wanting more, if possible elaborate a bit on what this means.*

    **[Response]** Based on this comment, we now write: "This approach successfully keeps the low frequency signal of the two original reanalysis products, yet provides additional high frequency signals which were lacking in previous products but are essential for investigating extreme events.".

*(9) L78: Elaborate on what Tair is, is it 2m air temperature? For translation to ESMs this is quite important.*

    **[Response]** Yes, Tair is 2 m air temperature, and this is now made clear in the revised manuscript version.

*(10) L80: Does the author perform regridding to 2.5°x2.5° so that these variables are on the same grid with each other as well as ESMs used? This should be elaborated on and if not done, justified.*

    **[Response]** Yes, we now clarify the re-gridding procedure, writing (L118-L121 in the Method): "Since precipitation is also not only controlled by local temperature (Zhou et al., 2020), we seek links to features of Tair from all grid cells. There are 10,368 gridboxes considered, at a common 2.5° × 2.5° resolution, that we use to for both historical observations and future ESMs, after remapping the latter. It is this Tair data and model simulations that we use across the globe to predict precipitation variation.".

*(11) L92: Out of curiosity, why first-order? Other archives (e.g. Brunner L., M. Hauser, R. Lorenz, and U. Beyerle (2020).The ETH Zurich CMIP6 next generation archive: technical documentation. DOI: 10.5281/zenodo.3734128) use second order.*

**[Response]** In practice, it has little differences between first-order and second-order conservative remapping technique for ESMs' Tair or precipitation resampling. Of course, the higher order conservative resampling method could lead higher level of "conservative". Considering the error or uncertainty in gridded precipitation data, first-order conservative sampling method is enough. We also noticed that many research use the first-order conservative re-gridding method for precipitation or Tair (e.g. Na et al., 2020, DOI: 10.1029/2019JD031926; Samset et al., 2018, DOI: 10.1002/2017GL076079). We now clarify in the Method: "As there remains substantial error and uncertainty in gridded precipitation data, we retain only a first-order re-gridding method for precipitation or Tair (we note second order calculations have been used by others [Brunner et al., 2020])".

*(12) L94: When calibrating, do you use all initial condition ensemble member available per ESM for SSP 5-8.5? Same goes for testing on the other SSPs?.*

   **[Response]** No, we used only one initial condition ensemble member for each ESM, most of which are 'r1i1p1f1', except ACCESS (r10i1p1f1), CanESM5 (r11i1p1f1) and UKESM (r1i1p1f2). This is because these three ESMs lack the initial conditions for 'r1i1p1f1'. We have clarified this in Table 1.

*(13) L94: Instead of going straight into calibration the author should also first provide the PREMU framework, consider therefore breaking up the text to introduce general approach (gridded temperature, PCR etc. ) , emulator framework (here introduce equations and key variables alongside Table 2 and Figure 1), calibration etc.*

   **[Response]** Thanks for the suggestion, which we have followed. We have divided the "Method" into "General Approach" and "Framework of PREMU" and we introduce the "Calibration" and "Validation" in the "Framework of PREMU". These are substantial changes, so we have not repeated them below – we leave these as direct edits and rearrangements of the revised manuscript.

*(14) L113: It is not convincing that taking a 3-month average across the month of interest and the preceding 2 months will sufficiently capture the influence from the previous 2 months. Moreover this could average out the general seasonal transitions in temperature, does the author have a justification for this instead of using e.g. a multiple linear regression.*

   **[Response]** Please see our full response to a similar general comment 2 above.

*(15) L113-L115: "ESM may under-represent the effects of topography and aerosols on the precipitation from observations" do you have a reference to back this up with? I see this later on in the discussion, consider bringing them up here too..*

   **[Response]** Thanks. We have cited two references here: "In addition, considering that ESMs may under-represent the effects of topography and aerosols on modelled precipitation (Samset et al., 2016; Yang et al., 2021), we calibrated the emulator for the historical period and future separately."

   Samset et al.: Fast and slow precipitation responses to individual climate forcers: A PDRMIP multimodel study, Geophys Res. Lett., 43, 2782-2791, https://doi.org/10.1002/2016GL068064, 2016.

Yang et al.: An evaluation of CMIP5 precipitation simulations using ground observations over ten river basins in China, Hydrol. Res., 52, 676-698, 10.2166/nh.2021.151, 2021.

*(16) L120: How the spatial patterns of the PCA components is obtained needs to be elaborated upon*

**[Response]** It has been revised as "The PCA component coefficients, $U_{e,m}^{Spatial}(k,i)$, are the combination of eigenvectors of the covariance matrix of the anomalies of the gridded 3-month average Tair."

*(17) Fig 2: It is strange that there is such consistently high coefficient values occurring in the Arctic, is there a reason to this? If possible, the explanatory power of such spatial patterns towards precipitation should somehow be evaluated in terms of how representative they are of global circulation patterns and inter-annual modes of variability. In such, one suggestion could be to check how the coefficients change when fitted using a leave-one-year out approach.*

**[Response]** First, we checked robustness using the leave-one-year out method to evaluate the sensitive of $U_{e,m}^{Spatial}$ to sample size. Taking the coefficients in July as an example (Fig. R1), we find that the standard deviations of coefficients themselves are quite small, which confirms the coefficient values to be stable. For the inter-annual modes of variability, Fig. R2 shows each climate index has a correlation coefficient >0.4 or <-0.4 with at least two PCs. Further the 15 climate indices can be well reconstructed by multi-linear regression by using the ten PCs as independent variables ($R^2$ = 0.40 – 0.94), which again implies that the combinations of the top ten PCs have high explanatory power for the global circulation patterns and inter-annual modes of variability.

In PCA, the larger the variance explained by a principal component, the more important that component is, and so the coefficients of the top ten PCs are larger in the regions with higher Tair variations. This is the reason for the high coefficient values in the Arctic and Antarctic in Fig. 2 and Fig. S2 (Fig. R8). The result that temperature variation is higher near the poles is well-established in the climate literature.

[Figure]

**Figure R8 a)** The standard deviations of Tair in a) January and b) July during the period 1901-1950.

*(18) L141: I find equation 3 redundant as it is the same as equation 2 only with grid-point specific coefficients, for simplicity's sakes perhaps these two equations could be merged.*
   **[Response]** Thanks for the suggestion. These two equations have been merged.

*(19) L142-L143: Redundant as the same as before but only for grid-point level.*
   **[Response]** Thanks. We now write: "To provide the reliable estimation of global monthly precipitation $P_{e,m}^{\widehat{global}}(y)$, we constructed the regression relationship between GLAP and the 10 individual principal components individually, following Eq. (2).".

*(20) L153, L156, L157: These equations are repetitive, again consider simply referencing to the ones already introduced before.*
   **[Response]** Thanks. We have deleted Eqs. (4-6).

*(21) Validation: Generally this section cuts off to abruptly and should be restructured as well as more detailed. There seems to be too much emphasis on defining the predicted variables and how they are obtained from the test set such that the actual method for validation is lost. The author should try be more explicit on what they are validating for i.e.*
*1.   Validation is performed for each predicted variable, $T^{PCA}$, $P^{global}$, $P^{grid}$*
*2.   What is validated for? It should be specified that the MAP, the inter annual variance and the trend are checked etc. Evaluation metrics should be provided (e.g. percentage error, correlation) and how they are calculated should also be explained.*
*3.   Given the high spatio-temporal variability of precipitation, it would be interesting to see how well PREMU does in preserving spatio/temporal autocorrelations, e.g. lag-1 correlations between months and spatial cross-correlations*

*4. Readers are left wondering if any evaluation on the PCA is done e.g. comparison between ESMs and observations, linkage to existing literature. There should be some physical relevance to the PCA components obtained as well as stability checks e.g. is 70% of variance explained by 10 PCA components within ESMs too, does this apply for all months, what happens to the spatial patterns when some training years are left out.*

**[Response]** Many thanks for the detailed comments. We have revised the "Validation" section following the referee's suggestions, as follows. 1) We introduced a discussion of the predicted variables directly in to the "Validation" section; 2) We added the new subsection "Evaluating PREMU" to introduce the detailed evaluation metrics in the revised version. That new subsection is: "In order to evaluate the advantages of PREMU compared to the emulations of gridded precipitation by other emulators (e.g. IMOGEN-based or OSCAR-based method), we simply used a prediction that linearly relates rainfall changes to global temperature changes and variability. We compared this simpler linear approach with the performance of PREMU. To evaluate how well the performance of PREMU at describing the historical observations, we compared the MAP and trends of GLAP from GSWP3 with the emulated values obtained from PREMU and the simple linear approach (subsection Sect. 4.1). Our statistic to compare was the Pearson correlation coefficients between the GLAP from observations and these two emulations. For gridded precipitation, we instead calculated the percentage error of MAP during 1987-2016 and compared the changes of MAP in the first and last three decades of the validation period (1951-1980 and 1987-2016) for each grid. Similarly, we evaluated the performance of PREMU at describing future precipitation by comparing the MAP, and trends of GLAP with ESMs, for the four future scenarios (subsection Sect. 4.2). As PREMU is an ESM-specific emulator, we calculated the performance of PREMU on each ESM individually. The percentage error at each grid was used to evaluate the emulations of PREMU at different locations. We additionally evaluated the performance of PREMU on the seasonal cycle of precipitation (subsections Sect. 4.3.) Here, we compared the land average precipitation in different latitudes from GSWP3 or multi-model mean of 9 ESMs with the emulation of PREMU. Also, the error in spatial pattern of seasonal precipitation is used to show that PREMU can capture the seasonal cycle of gridded precipitation from both historical observations or ESMs."; 3) We agree that it is necessary to evaluate whether PREMU can preserve spatio/temporal autocorrelations. We detrend/de-seasonal the residual series of monthly precipitation by applying the Singular Spectrum Analysis method. The result shows a significant but low month-to-month $lag_1$ correlation of the residual series from GSWP3 ($lag_1$ correlation = 0.05, p<0.05). We did find that PREMU overestimated the $lag_1$ correlations ($lag_1$ correlation = 0.16, p<0.05). The potential physical explanations of this lag correlation need further study, which is out scope of this study.

4) To answer point 4 by the reviewer, we first note that more than 70% of variance can be explained by the top 10 PCs within ESMs (Fig. S3a and Fig. S4a.). After applying the leave-one-year out method, the spatial pattern of coefficients is stable with a low standard derivation for all PCs (Fig. R1). In addition, we have evaluated the stability of the spatial part of PCA between different emission scenarios in Sect. 5, (as Fig. S13 and S14; now Fig. S19 and S20). We now write: "Furthermore, we evaluated the reliability of our assumption of constant spatial part of PCAs in Sect. 5. To test this assumption, we compared the coefficient matrices of temperature derived for the SSP5-8.5 scenario with those from the SSP1-2.6 scenario. Finally, for the further developments of PREMU, we explored the performance of emulating precipitation by other versions of

PREMU (e.g., PREMU-1mon, PREMU-6mon and PREMU-land), and again our findings are presented in Sect. 5."

*(22) L162, Eq 7: This is a methodological addition and seems out of place in the validation section, consider moving it to methods (e.g. "from further tests etc. we decided to apply a correction factor etc.").*

**[Response]** We agree. We have moved it to the new section titled "Evaluating PREMU".

*(23) L170: when taking the areally-average is it a weighted average according to latitude, if not why? Areally-averaged could also be replaced by simply average or average over grid points as areally-averaged reads ambiguously.*

**[Response]** Yes, we clarify this - we used the weighted average according to latitude and we revised the wording of "area-weighted average" to instead be "average over grid points as area-averaged".

*(24) L172: How do you check the similarities in inter annual variations i.e. using pearson correlation coefficient between the inter-annual variance or absolute values or applying a low-pass filter to extract the variance before checking the correlations?*

**[Response]** Here, we used the Pearson correlation coefficient between the inter-annual variance directly and we added a description in the revised manuscript "we compared the MAP and trends of GLAP from GSWP3 with the emulated values obtained from PREMU and the simple linear approach (subsection Sect. 4.1). Our statistic to compare was the Pearson correlation coefficients between the GLAP from observations and these two emulations." We also calculated the Pearson correlation coefficient after applying a low-pass filter, e.g., linearly detrend. By applying a low-pass filter, the correlation coefficient between GLAP in validation period from PREMU and GSWP3 (R=0.66) are close to the coefficient without low-pass filter (R=0.67). However, the coefficient of GLAP between Simple Linear method and GSWP3 are smaller after applying the low-pass filter (R=0.15 without low-pass filter; R=-0.03 with low-pass filter).

*(25) Results: Quite nice findings! It should be made explicit the annual precipitation is mainly verified for, given that PREMU is a monthly emulator. It raises the question of the seasonal performance of PREMU, is there a reason the authors did not show this?*

**[Response]** Thank you for your positive evaluation of our results. In the revised version, we have added a subsection that describes the performance in describing the seasonal cycle (Sect 4.3): "As a monthly emulator, the performance of PREMU at describing the seasonal cycle of precipitation requires evaluation. For the historical observations, PREMU can capture the seasonal cycle of GLAP in each latitude band (Fig. 11). There are some spatial differences, with little error in boreal regions but with 18-22 mm mon$^{-1}$ error in tropics (Fig. S16). We found fewer grid cells showing more than 20 mm mon$^{-1}$ error of seasonal precipitation from GSWP3 in our emulation (18% - 24%) compared to using a simple linear fit (24% - 31%; Fig. S16). As for the future precipitation, PREMU shows a good performance on emulating the monthly land average precipitation in each latitude band under all scenarios (Fig. S17). However, PERMU tends to overestimate the JJA (June, July, August) and SON (September, October, November) precipitation in South Asia and Amazon and underestimates the SON and DJF (December, January, February) precipitation in West Africa under SSP1-2.6, while the error in spatial pattern of seasonal

precipitation decreases from low to high emission scenarios (Fig. S18).". See response to general comments 3.

*(26) L271-L305: These seem to be more points within the results, in general it would be good to first introduce that these things (e.g. memory effect) were tested for an how in the methods and then provide the results in the Results. Otherwise the Discussion is lacking structure and quite long.*

> **[Response]** After careful consideration, we have decided to leave these three paragraphs in the discussion. We fear that moving these into Methods and Results may confuse readers of the main headline point of manuscript, which is that the 3-month PREMU emulator performs well in describing the gridded precipitation from both observations and ESMs. However, we do understand your concerns that Discussion is lacking structure. To make the Discussion clearer, we split in to specific subsections of: "Possible cause for the emulation errors of PREMU", "Evaluating the assumptions in methods", "Other versions of PREMU" and "Potential further developments of PREMU".

*(27) L311: Different version of calibrated PREMUs should probably be introduced in their own subsection*

> **[Response]** Thanks. In the revised version, we separated in to two individual subsections: "PREMU constructed by different temperature lag periods" and "PREMU constructed by Tair over land".

*(28) Conclusion: Generally the discussion is quite long and covers myriad of possibilities, consider merging some parts into the results (as mentioned above) some parts in conclusion and dividing into further subsections e.g. Further Developments/Versions (PREMU-1MON, PREMU-LAND etc.)*

> **[Response]** After careful consideration, we request to leave the broad manuscript structure unchanged. However, as noted above, we do now split the Discussion into four subsections to make the Discussion clearer.

*(29) Table 3: It is hard to contrast so many numbers, perhaps providing percentage difference relative to actual trend value or something along those lines would be better?*

> **[Response]** After consideration, we prefer to keep Table 3 (now the Table 4) in the manuscript. This is because Table 3 not only provides the errors of trends of GLAP in PREMU, but interestingly also shows a higher trend of GLAP in CESM2 and GFDL-ESM4 under scenario SSP2-4.5 than SSP3-7.0. We have discussed this in Discussion, that this may be caused by the more substantial land use changes of SSP3-7.0, and that are accounted for in ESM simulations. Such land use may result in a slower increase in precipitation.
>
> However, following this suggestion, we have added a new Table in supplementary information (Table S1) that shows the percentage difference relative value.

*(30) Figure 1: Having read the text I understand this figure, however by itself it is quite complicated and could benefit from some more graphics and simplification of the text.*

**[Response]** We agree. We reduced the descriptions of text and added subtitles for figures to make Figure 1 easier to understand.

*(31) Figure 4: Readers would be interested in seeing differences in trends between GWSP and PREMU characterised, it seems like there are areas where the direction in trend is different (e.g. Australia, West Africa) which also matters vs simply looking at over/underestimation of changes. The discussion point that more differences can be seen in this figure as compared to Figure 8 due to topography and aerosols should also be mentioned when describing this figure as it enriches the analysis.*

**[Response]** Thank you for the suggestion. We added some descriptions about the different directions of trends in some regions (e.g. Australia, West Africa) and its possible reason in the revised version: "Furthermore, we noted that the changes in MAP from both PREMU and the simple linear method show the opposite to the changes in MAP from GSWP3 in some regions (e.g., Australia and West Africa; Fig. 4b, d, f). This suggests that changes in precipitation in these particular regions may be driven by the factors such as aerosols, topography and land use changes rather than temperature.". In addition, some descriptions also be added for Fig. 8: "Compared to the historical period, PREMU can capture the changes in MAP in west Africa well under all four scenarios, but also emulated the opposite changes of MAP in Australia under SSP3-7.0, which will be discussed in Sect. 5". And also, in Disscussion: "Considering the changes in MAP in west Africa from PREMU are opposite to the changes in MAP from GSWP3 (Fig. 4), PREMU can emulate the changes in MAP in west Africa from ESMs well.".

*(32) Figure 8: again it may be interesting to characterise the differences: where direction of changes are properly captured and where they are opposite*

**[Response]** Yes, as mentioned above, we have added some descriptions for Fig. 8: "Compared to the historical period, PREMU can capture the changes in MAP in west Africa well under all four scenarios, but also emulated the opposite changes of MAP in Australia under SSP3-7.0.".

**Editorial Comments:**

*(33) L28: I would start the sentence off with the subject rather than verb, especially if it is at the beginning of the paragraph, i.e.*
*Earth system models (ESMs) are the primary tools to study the impact of greenhouse gas emissions on our climate, representing all the important Earth system processes (IPCC, 2013).*

**[Response]** Thanks. It has been revised, using the suggested wording.

*(34) L30: run **the***

**[Response]** It has been revised.

*(35) Figure captions: emulations are referred to as **our** emulations, I am not sure how formal this is.*

**[Response]** Thanks, we changed the "our emulations" to "emulations from PREMU" in the figure captions in the revised manuscript.

*(36) Section 2.3.2: the Table should be labelled and given a number*
**[Response]** Sorry, this typo has been corrected, and it has been labelled as "Table 3".

*(37) Eq 4: T has the superscript Timeseries,val which is not in line with that of eq 1, is this intentional?*
**[Response]** Actually, the superscript "val" of T in Eq.4 means the "validation datasets". We have, though, now delated the Eq.4 in the revision manuscript. (If not, we would change $T_{e,m}^{Timseries}(k,y)$ to $T_{e,m}^{Timseries,cal}(k,y)$ in Eq.1.)

**References**

Dai, A.: The influence of the inter-decadal Pacific oscillation on US precipitation during 1923–2010, Clim. Dynam., 41, 633-646, 10.1007/s00382-012-1446-5, 2013.

Li, W., Zhai, P., and Cai, J.: Research on the Relationship of ENSO and the Frequency of Extreme Precipitation Events in China, Adv. Clim. Change. Res., 2, 101-107, https://doi.org/10.3724/SP.J.1248.2011.00101, 2011.

McKinnon, K. A. and Deser, C.: Internal Variability and Regional Climate Trends in an Observational Large Ensemble, J. Climate, 31, 6783-6802, 10.1175/JCLI-D-17-0901.1, 2018.

McKinnon, K. A. and Deser, C.: The Inherent Uncertainty of Precipitation Variability, Trends, and Extremes due to Internal Variability, with Implications for Western U.S. Water Resources, J. Climate, 34, 9605-9622, 10.1175/JCLI-D-21-0251.1, 2021.

Na, Y., Fu, Q., & Kodama, C. (2020). Precipitation probability and its future changes from a global cloud-resolving model and CMIP6 simulations. Journal of Geophysical Research: Atmospheres, 125, e2019JD031926, https://doi.org/10.1029/2019JD031926

Nath, S., Lejeune, Q., Beusch, L., Seneviratne, S. I., and Schleussner, C.-F.: MESMER-M: an Earth system model emulator for spatially resolved monthly temperature, Earth Syst. Dynam., 13, 851-877, https://doi.org/10.5194/esd-13-851-2022, 2022.

Samset, B. H., Myhre, G., Forster, P. M., Hodnebrog, Ø., Andrews, T., Faluvegi, G., Fläschner, D., Kasoar, M., Kharin, V., Kirkevåg, A., Lamarque, J.-F., Olivié, D., Richardson, T., Shindell, D., Shine, K. P., Takemura, T., and Voulgarakis, A.: Fast and slow precipitation responses to individual climate forcers: A PDRMIP multimodel study, Geophys Res. Lett., 43, 2782-2791, https://doi.org/10.1002/2016GL068064, 2016.

Samset, B. H., Sand, M., Smith, C. J., Bauer, S. E., Forster, P. M., Fuglestvedt, J. S., Osprey, S., & Schleussner, C.-F. Climate impacts from a removal of anthropogenic aerosol emissions. Geophysical Research Letters, 45, 1020– 1029. https://doi.org/10.1002/2017GL076079, 2018

Tebaldi, C. and Arblaster, J. M.: Pattern scaling: Its strengths and limitations, and an update on the latest model simulations, Clim. Change, 122, 459-471, 10.1007/s10584-013-1032-9, 2014.

Tebaldi, C. and Knutti, R.: Evaluating the accuracy of climate change pattern emulation for low warming targets, Environ. Res. Lett., 13, 055006, 10.1088/1748-9326/aabef2, 2018.

Tsanis, I. and Tapoglou, E.: Winter North Atlantic Oscillation impact on European precipitation and drought under climate change, Theor. Appl. Climatol., 135, 323-330, 10.1007/s00704-018-2379-7, 2019.

Yang, X., Yong, B., Yu, Z., and Zhang, Y.: An evaluation of CMIP5 precipitation simulations using ground observations over ten river basins in China, Hydrol. Res., 52, 676-698, 10.2166/nh.2021.151, 2021.

**2. Response to Comments from Reviewer #2:**

*The authors present a novel method for precipitation emulation (PREMU) that uses gridded temperature patterns over time to (re)construct global and local precipitation time series by means of principle component analysis (PCA). The method provides high accuracy against existing precipitation estimates, suggesting its potential power for situations where a temperature projection can be easily constructed but a precipitation projection cannot, such as with lower complexity models (LCMs) or novel projections of temperature outside of the most common future scenarios in more complex earth system models (ESMs). The organization of the text does not make this message as clearly as it needs to, though, and there are a few outstanding questions about the methodology as well that need elaboration.*

*The effort put into defending the ability of PREMU to accurately reconstruct ESM precipitation overwhelms any description of the applicability of the method to LCM temperature data, making that reproducibility appear to be the main message instead. While the accuracy of PREMU in this respect is impressive, it minimizes the importance of this novel method because it gives the impression that no new precipitation has been added. More emphasis on the potential application of PREMU is needed, and better framing of the ESM matching would greatly improve the article as well.*

*In particular, with PREMU being much less computationally intense than an ESM simulation, this paper should include at least one application of PREMU to an LCM projection of future temperature as an example. The example should also include a precipitation simulation based on that LCM using the traditional linear method so that the audience can see what differences (in theory, improvements) there are between the linear method and PREMU.*

*Some specific suggestions for providing better framework and organization for the core message are presented below, along with questions and comments about the methodology. Overall, I do believe that this is a valuable new tool for the community that should be shared; it just needs slightly more explanation and a better presentation.*

> **[Response]** We thank the reviewer for the positive feedback and constructive comments/suggestions. We are pleased that the approach is deemed "novel", and in particular that PREMU is likely to be a "valuable tool for the community".
>
> Following the comments/suggestions, we have carefully revised the manuscript. First, we have re-structured the Introduction, split Data and Method, and added subsections for Methods and Discussion. Second, we point ahead to comments that will be made in the Discussion section after each choice we made in Method. As for coupling with other LCMs, we agree this is the most important eventual application of PREMU. Before evaluating the results of coupling PREMU with other LCMs, we need to make sure that PREMU can reproduce the characteristics of precipitation from both observations and ESMs. Hence much of this paper is about emulator verification, and that is reflected in our choice of journal.
>
> Following the reviewer's suggestion, we have tried to couple MESMER and MESMER-M (a LCM that can simulate land temperature) with PREMU-land, calibrated with CESM2 outputs of Tair under SSP5-8.5 (we need to select an ESM, as these three emulators are

all ESM-specific). The results and details are shown in the response to Comment #21 below. Note that PREMU emulated precipitation using gridded land and ocean temperature is better than that only using gridded land temperature that MESMER and MESMER-M emulated. A formal coupling of a LCM and PREMU is complex, and possibly goes beyond the bounds (and message) of this manuscript, which already contains >10 figures. However, we do want to acknowledge the reviewer's point about potential future applications, including as a way to extend the usefulness of LCMs. Hence, we now write in the Conclusion: "To the best of our knowledge, this is a pioneer emulator that can be directly coupled within existing LCMs, and especially noting that LCMs may perform well for other variables but are currently weaker at estimating features of rainfall. This new combination will better predict global and gridded precipitation under different emission scenarios.".

**Specific Comments:**

*(1) Abstract: The mention of LCMs should be mentioned much earlier, along with the issue with linear scaling for precipitation. Then present PREMU as a solution to that issue, followed by the justification statistics from the historical and ESM comparisons. L22 onward can remain as the conclusion sentence. L12 "better estimate and represent precipitation simulated by Earth system models (ESMs)" wording should be changed as there is no justification in the paper suggesting the PREMU simulations are better than the ESM data they are recreating.*

    **[Response]** In the revised manuscript, we have mentioned LCMs earlier to introduce the issues in emulating precipitation by simple linear scaling methods: "Some current lower complexity models (LCMs) are designed to emulate precipitation in a computationally effective way. However, for precipitation in particular, they are known to have large errors, due to their simpler linear scaling of precipitation changes against global warming (e.g., IMOGEN; Zelazowski et al., 2018).". Also, L12 has been changed to make clear that PERMU can emulate ESMs (it is not designed to out-perform ESMs). We now write: "estimate and represent precipitation well, as simulated by different Earth system models (ESMs) and under different user-prescribed emission scenarios."

*(2) Introduction: Generally good, though precipitation is not mentioned until L47. This section should be rearranged to better emphasize the problems associated with precipitation estimation outside of the common ESM future scenarios; it does not have to lead the section but should be mentioned higher up and expounded upon. The final paragraph also needs to be fleshed out more, particularly in the sentence describing section 4 where you need to draw the connection between the presented ESM validation and the potential use with LCMs. The potential addition of a direct LCM-PREMU precipitation example would also be mentioned in these last two sentences.*

    **[Response]** Following the suggestions, we have restructured the Introduction. First, we merged the first two paragraphs to clearly represent the background of this research. Then, we now highlight earlier the message that precipitation is a crucial variable but it has statistical features that make its representation within traditional LCM approaches difficult. Hence at the beginning of next paragraph, we state: "Precipitation has high spatio-temporal variability and is affected by atmospheric dynamics, inter-annual modes of variability (Li et al., 2021; Tsanis and Tapoglou, 2019), making representing within tradition al LCM approaches difficult." In

addition, we added the description of potential implication of coupling PREMU with other LCMs in the last paragraph of Introduction: "We have designed the presented emulator such that it can act as, for instance, an enhanced precipitation module for the OSCAR and IMOGEN models. Alternatively, PREMU could be coupled directly with the MAGICC-MESMER and MESMER-M emulators."

*(3) Data & Methods: With the existing structure in this section, it appears that the Methods might benefit from being its own top-level section, as Methods is the only section of the paper with third-level subsections. There are also many choices (listed in detail below) made in the methodology that are ultimately justified in the Discussion section but present lingering questions when presented alone in this section. Full justification of these choices cannot be done before the Results section, but you should telegraph some of the comments that will be made in the Discussion section so that your audience recognizes you are already aware of some of the potential issues with your assumptions and choices.*

> **[Response]** Thank you for the suggestions. We have split Data and Methods into two independent sections in the revised version. For the selection of the 3-month average, we have point forward to discussions in Sect. 5: "The choice of the 3-month average Tair as independent variables is robust, with further details illustrated in Sect. 5.". We have added the description of reliability of PCA in the last paragraph of Methods and also mentioned this will be evaluated further in Sect. 5, writing: "Furthermore, we evaluated the reliability of our assumption of constant spatial part of PCAs in this study in Sect. 5.".

*(4) L89-90: Calibrating to your endpoint (here, the hottest future temperature scenario) is good in the sense that your validation will only involve interpolation instead of extrapolation, but since we know that the atmosphere can respond nonlinearly with warming, this introduces the concern that this extreme warming scenario may not produce precipitation patterns that are representative of cooler scenarios. This is addressed later (e.g. L203-205, L271-292) but should be acknowledged here.*

> **[Response]** We agree that it's better to mention the potential worse performance on the cooler scenarios due to nonlinearly respondence to warming. We have added this in the revised manuscript: "But note that the emulator based on this extreme warming scenario (SSP5-8.5) may not produce well the precipitation patterns of cooler scenarios (e.g., SSP1-2.6) due to the nonlinear response of the atmosphere to warming. We discuss this further in Sect. 5."

*(5) L91 "we constructed the emulator for each ESM respectively" On first reading this, it sounds like you performed the entire PCA individually for ESM. Upon reading further, it instead looks like only the coefficients differ for each ESM, as Fig. S2+S3 state that the same ten temperature modes are used for all ESMs. This should be stated clearer*

> **[Response]** Sorry for the misleading description. Actually, we did construct the full emulator for each ESM respectively (so not just the PCA coefficients). Hence the corresponding $U_{e,m}^{Spatial}$ is derived for each ESM respectively. We have revised the figure captions of Fig. S2, S3 (now Fig. S3, S4) as: "The average of coefficients of the top ten principal components across the nine ESMs."

*(6) L95 Calibration: This is another section that should be split in half, as most of the first half of the section (L96-177?) focuses on the temperature PCA itself. This description should also use a little bit more elaboration for readers who may not be as familiar with the mathematics of the process.*

**[Response]** Following this suggestion (also raised by the reviewer #1) we have split the "Method" into three subsections: General approach, Framework for PREMU and Evaluating PREMU. Then, we introduced the temperature PCA method in the General approach. We have also added serval references which applied PCA method in climate research, for the readers who may not be familiar with the PCA: "As a method that can be used to overcome the problem of multilinearity on predictor variables, the PCR technique is widely used in forecasting seasonal precipitation (Kim et al.,2017b) and reconstructing the climatic modes of variability (Jones et al., 2009; Michel et al., 2020). These authors find PCA method to perform remarkably well, and the method is explained in detail in Chapter 13 of Storch and Zwiers (1999).".

*(7) L107 "used in climate research" requires at least two example references.*

**[Response]** We have added three references here in the revised version: "We followed the standard procedure of Principal Component Analysis (PCA), as used in climate research (e.g., Yan et al., 2020; Singh, 2006; Jiang et al., 2020), that consists of a set of time-invariant spatial patterns multiplied by timeseries of coefficients.".

Yan et al., 2020: Eastward shift and extension of ENSO-induced tropical precipitation anomalies under global warming.

Singh, 2006: Pattern characteristics of Indian monsoon rainfall using principal component analysis (PCA).

Jiang et al., 2020: Principal Component Analysis for Extremes and Application to U.S. Precipitation.

*(8) L111-112: You further investigate the effect of different lags later in the text, but this statement alone requires some elaboration and at least one reference to justify how earlier months must be considered.*

**[Response]** These references (Li et al., 2011; Lu et al., 2019; Efthymiadis et al.,2007) have been added: "As precipitation may be influenced by the temperature of the previous months via large scale circulation (e.g., the lag effects of ENSO on regional precipitation; Li et al., 2011; Lu et al., 2019; Efthymiadis et al., 2007)".

*(9) L116-117: something like this should have been mentioned around L89-90, and similar statements (i.e. "discussed in Sect. 4") are needed for other issues.*

**[Response]** OK. We have delated this sentence because the similar statement appears around L89-90 (now L102-104).

*(10) L118: It's great to have the table for reference, but more of this information should also be in the text itself so readers don't have to go back and forth between the table and the text repeatedly in order to understand the equations.*

**[Response]** OK. We have also introduced in the main text Eqs. 1 and Eq. 2 (the PCA structure for temperature and precipitation). Further, we will request that the journal typesetters place the Table on the same page, to support ease of reference.

*(11) L127-128: More justification is needed as to why both the historical and ESM versions of PREMU were constructed with 10 modes of temperature. From the four figures cited here, in addition to S13+S14, it's clear that the amount of warming in the chosen temperature data greatly affects how many modes are needed to hit any particular threshold for variance explained. This also goes back to the earlier concern of how the 8.5 scenario, which is so strongly dictated by worldwide warming, might not give reasonable precipitation results for scenarios with more varied temperature modes. This is later somewhat addressed in L275-278, but the stark differences between the first four figures mentioned in L128 will raise questions that should be partially addressed here.*

**[Response]** We find that different emission scenarios require a different number of modes to hit any particular threshold for variance explained. Hence choosing the top 10 PCs as independent variables is, arguable, an arbitrary choice, but we find it does achieve high variance explained in all cases. As different PCs may represent different characteristic of gridded Tair (e.g., $PC_1$ represents the increase trend of Tair and $PC_2$ - $PC_{10}$ may represent different inter-annual variations of Tair; Fig. R1), the number of PCs required may also be decided by the research purpose. For example, if the research only requires the decadal average or increase trend of precipitation, PREMU calibrated by the top 1 PC of Tair under SSP5-8.5 (PREMU- one PC) can capture the trends of GLAP under all four scenarios (Fig. R2, also shown as Fig. S21). For these reasons, and triggered by this reviewer request, we have added a statement about the choice of the number of PCs in Sect. 5: "Depending on circumstance, we noted that it may not be necessary to use all top ten PCs. For instance, if researchers only require the decadal average or any increasing trend in precipitation, PREMU calibrated by the top 1 PC of Tair under SSP5-8.5 may be sufficient to capture these characteristics of GLAP from ESMs and under all four scenarios (Fig. S21)." Nevertheless, we recommended that using more PCs may lead to better emulations of precipitation. Following the suggestion, we indicate that we would discuss the stability of the $U_{e,m}^{Spatial}(k, i)$ of the top 10 PCs in Sect. **5**: "The stability of the component coefficients $U_{e,m}^{Spatial}(k, i)$ of the top 10 PCs, between different scenarios, is discussed in Sect. 5.".

[Figure]

**Figure R1** The average of top 10 PCs of gridded January Tair under SSP1-2.6 and SSP5-8.5 across nine ESMs.

[Figure]

**Figure R2** The emulation of PREMU based on only the top single PC of future precipitation:
**a**) multi-model mean GLAP in 9 ESMs from CMIP6 ("CMIP6(9)"), and the precipitation
prediction by our emulator (PREMU- one PC) in 2015-2100 under the SSP5-8.5 scenario.
The shaded area represents the mean±std. b) The spatial pattern of error in MAP during
2071-2100 between multi-model mean and our emulator. The emulation of precipitation
under **c-d**) SSP1-2.6; **e-f**) SSP2-4.5; **g-h**) SSP3-7.0 by the PREMU- one PC calibrated under
SSP5-8.5.

*(12) L136,L141: The similarities between these two equations suggest that it should be possible
to construct the global coefficients from the gridded ones, i.e. instead of doing the global
analysis alone. I know some averages are computed and compared later, but unsure if that is
the same as what I'm thinking here.*

*L159-164: if you say you found a slight difference, you should be more explicit as to what the difference, and potentially offer an explanation for why the difference is present. Without a physical reason given, this subsequent ratio correction feels like a poorly justified mathematical band-aid.*

**[Response]** Yes, the emulation of GLAP and area-weighted average of the gridded precipitation are theoretically equal based on these two equations. However, we found that a numerical issue the emulation of gridded precipitation, but with global calibration, can cause estimate of rainfall that are is less than zero in some months. We can reset these values equal to zero, which will bring slightly difference (less than 1 mm yr$^{-1}$) between GLAP and area-weighted average of gridded precipitation. Following the suggestion, we added: "brought by setting negative numbers of $\widehat{P_{e,m}^{grid}}$ to be zero at some grid points.".

*(13) Results: This section seems thorough, but it often states a large number of different quantities in quick succession in many paragraphs, making it somewhat difficult to follow and occasionally feel like not every comparison is being directly stated. I would like to see a more organized pattern of describing results in each paragraph and/or another table stating all values of interest (average precip/yr, precip trend, year to year variance, error, correlation, proportion of grid cells with given error, etc.) for each set of simulations, both trainings and experiments, obs vs linear vs PREMU for historic and each ESM vs PREMU for future.*

**[Response]** Thanks. We agree it is helpful to add a table for presenting all values of interest. We have added two tables in the supplementary (Table S1 and Table S2).New Table S1 shows the comparison of characteristics (e.g., GLAP; trend of GLAP; correlation coefficients; proportion of grid cells with more than 25% error of MAP) of GLAP and gridded precipitation between observations and emulations from Simple Linear and PREMU, and Table S2 presents the comparison of characters between ESMs and the emulations.

*(14) Fig. 3: correlation and error look like they might be strongly influenced by the extreme changes in the last year or two of the experiment time period. Out of curiosity, since PREMU is not too computationally intensive, could you do some sensitivity analysis where you vary the start and end years for both your trainings and experiments? I'm guessing it might not change much for your results and thus might not need to be put into a later version of this article, but it would be interesting to see.*

**[Response]** We tested this sensitivity analysis for calibration PREMU by the Tair and precipitation in 1903-1948 and validation with Tair and precipitation in 1953-2014 (Fig. R3). The correlations with GSWP3 and PREMU do not change much for both calibration and validation periods (r=0.82 for calibration period, and r=0.67 for validation period). The correlation with Simple Linear method does not change too much (from 0.15 to 0.21 for validation period). In general, there is little difference between these two results.

[Figure]

**Figure R3** Same as Fig.3 but for calibration during period 1903-1948 and validation during period 1953-2014.

*(15) L190-191: It feels very weird to end a section saying a particular part of the simulation is inaccurate - why not try to compare it to the linear scaling? Is it worse than the linear method, and that's why the linear scaling isn't mentioned here?*

**[Response]** Actually, both PREMU and the linear method underestimate / overestimate the changes in annual precipitation in these regions. We have mentioned this in the revised version "For the differences between these two time periods, PREMU shows consistent changes of precipitation in northern Eurasia, North America and the central South America and when using GSWP3 data. However, the simple linear method has underestimated the increase or overestimated the decrease of precipitation in these regions (Fig. 4). In some regions of East Asia, Europe, Australia and South America, PREMU underestimates / overestimates the changes in annual precipitation (in range 50-200 mm year$^{-1}$), and these values represent little improvement over the simple linear method (Fig. 4)". Furthermore, we have also point forward in the paper (to Discussion, Sect 5) concerning the possible reason for the inaccurate emulation of changes in gridded precipitation. We write: "Furthermore, we noted that the changes in MAP from both PREMU and the simple linear method show the opposite to the changes in MAP from GSWP3 in some regions (e.g., Australia and West Africa; Fig. 4b, d, f). This suggests that changes in precipitation in these particular regions may be driven by the factors such as aerosols, topography and land use changes rather than temperature, which will be further discussed in Sect. 5.".

*(16) L194: This is the first time mentioning the possibility of PREMU working with novel trajectories for GHGs; something that should be mentioned earlier in the paper alongside the extra emphasis on using PREMU with LCM temperatures.*

**[Response]** Thank you for this comment, and we appreciate being reminded of the purpose of emulators of more complex models. Hence we now state more clearly the potential applications of PREMU in Introduction of the revised version, writing: "We have designed the presented emulator such that it can act as, for instance, an enhanced precipitation module for the OSCAR and IMOGEN models. Alternatively, PREMU could be coupled directly with the MAGICC-MESMER and MESMER-M emulators.".

*(17) L201-202: While you later suggest that some of better variation simulation with ESMs might be due to topographic complication or aerosols, (L243-L252,) one thing that comes to mind with R values is the standard deviations in the data sets you're comparing. I would like to see some mention of the underlying statistics here.*

**[Response]** We provide a table for all statistics values in the revised paper version (new Tables S1 and S2). Additionally, we mention the differences in standard deviations of GLAP from high to low emission scenarios, as: "In addition, the standard deviations of GLAP are underestimated by 2% in SSP5-8.5 and by 43% in SSP1-2.6 (Table. S2).".

*(18) L204-205: This speaks to the previously mentioned issue of training to the 8.5 scenario, and you do address it later in the Discussion section. Nothing necessarily needs to be added here if you properly acknowledge the issue earlier in the text.*

**[Response]** Thanks, it has been deleted. We have acknowledged this earlier, in the Method: "But note that the emulator based on this extreme warming scenario (SSP5-8.5) may not produce well the precipitation patterns of cooler scenarios (e.g., SSP1-2.6) due to the nonlinearly response of the atmosphere to warming.".

*(19) L207-209: The issue of over half of your ESMs doing visibly worse than the others deserves more than the passing comment here, especially as you've devoted a figure (6) to show it. There is a much larger discussion in L254-279, but if you don't say more here, you should at least mention that the discussion is coming later. (Again, telegraph your discussion points so that if a reader starts to question a methodological point or an odd result, they know you're aware of it and are planning to address that concern).*

**[Response]** We have telegraphed here that the reason for the different performance of PREMU on these ESMs will be discussed more fully in Sect. 5: "These differences could be partly related to the substantial uncertainties and different features affecting future precipitation projections in ESMs. These factors are discussed in Sect. 5".

*(20) L212-223: Another paragraph that feels disorderly with the amount of widely varying results strung together.*

**[Response]** We have restructured this paragraph according to the following issues: 1. Introducing the performance of PREMU on gridded MAP and changes of MAP from the multi-model mean of ESMs; 2. Showing the performance of PREMU at emulating the gridded precipitation for each ESM. 3. Concluding, critically, that the errors by PREMU are much smaller than the inter-ESM differences.

*(21) Discussion: This is a great consideration of all potential issues with the methodology and interesting components of the results previously shown. As stated before, several of these need to be telegraphed earlier in the paper so your audience isn't reading through your results with too many questions. It has now been over six pages since "LCMs" was previously mentioned, which is why the article so far feels like the ESM comparison was the main goal, as opposed to the actual potential for use with temperatures from LCMs and other novel scenarios. Also, your audience should not get to this point without knowing exactly how one of these other*

*experiments would be run, e.g. what would be the "training" precipitation data for an LCM experiment?*

**[Response]** Thank you for the overall positive feedback regarding our Discussion. We have telegraphed many points from earlier in the MS now, to the Discussion (see response to the above comments). We certainly agree that the implication of coupling with other LCMs is the most important goal for PREMU, and we emphasized this in both Introduction: "Alternatively, PREMU could be coupled directly with the MAGICC-MESMER and MESMER-M emulators. When coupling with other LCMs, tracing gridded precipitation under novel trajectories for GHGs will be possible.", and in Conclusion: "To the best of our knowledge, this is a pioneer emulator that can be directly coupled within existing LCMs, and especially noting that LCMs may perform well for other variables but are currently weaker at estimating features of rainfall.". Following the reviewer's suggestion, we have undertaken an experiment of coupling one LCM that emulates monthly land gridded temperature only (MESSER and MESSER-R) with PREMU-land. This experiment was calibrated with CESM2 outputs of Tair under SSP5-8.5, and so these three emulators are all ESM-specific. Then we emulated precipitation by MESSER-PREMU-land under other three scenarios (SSP1-2.6, SSP2-4.5 and SSP3-7.5). The results shows that the MESMER-PREMU-land can capture both trends and inter-annual variations under these scenarios (Fig. R4), while the Simple Linear method – almost by definition - cannot reproduce the inter-annual variations of GLAP. This suggests a promising prospect to use the MESMER-PREMU chain to emulate gridded precipitation under SSP scenarios. Note that emulated precipitation using gridded land+ocean temperature is better than that only using gridded land temperature that current LCMs emulated (e.g., MESSER).

Considering to undertake the coupling properly, substantial additional work is needed to link LCM and PREMU (develop ocean temperature emulator, calibrate and validate for CMIP6 ESMs, and probably simple application of coupled LCM-PREMU), and that our manuscript is already very long (and >10 figures), we prefer for now to not include these initial results of coupled LCM and PREMU. We would like to fully show the coupled LCM-PREMU in our follow-on study. Thank you for your understanding, and hope this is OK with you.

[Figure]

**Figure R4** GLAP from CESM2 and the emulations from MESMER-PREMU and Simple Linear Method under **a**) SSP1-2.6; **b**) SSP2-4.5; **c**) SSP3-7.0; **d**) SSP5-8.5

*(22) L254-279: These are all good suggestions for why some ESMs are better simulated with the PREMU method than others. However, it does raise the question: if we know different ESMs have different schemes relating atmospheric circulation to their precipitation, why was a set of 10 temperature modes chosen to use for all ESM experiments with the only differences being the subsequent coefficients, especially when we know the ESM temperature patterns themselves are not consistent between different ESMs to begin with? You could potentially better capture model-specific ENSOs and other such features this way.*

    **[Response]** We are sorry for any potential ambiguity. We did construct the emulator in full for each ESM respectively. Hence additional to the timeseries coefficients, we derive

the corresponding ESM-specific $U_{e,m}^{Spatial}$ spatial temperature patterns. As we mentioned above, related potentially misleading figure captions have been revised to make this point clear.

*(23) L275-276: This paragraph is a great way to address some of the previously mentioned concerns about training to a scenario with such extreme warming. This particular sentence, though, still doesn't seem entirely reasonable considering how different the 8.5 modes and 2.6 modes are. You do later state that the mode order is slightly shuffled, implying that you have clearly identified similar pairs of modes between the two scenarios; it would be nice to show a side-by-side comparison justifying to your audience that these two sets of modes are indeed similar enough to explain the >95% similarity in coefficients.*

**[Response]** Following the suggestions, we have revised Fig. S13 and S14 (now Fig. S19 and S20) to show the pairs of coefficients of top 10 PCs in January and July between SSP5-8.5 and SS1-2.6 after adjusting the mode orders. The similar pairs of modes between these two scenarios have been shown in Fig. R5 (also shown as Fig. S19 now). The ">95%" in this sentence [L275/6] means the top 10 PCs in both SSP5-8.5 can explain >95% spatial variations of gridded temperature. As the similarities of these two modes are not necessary to be quantitative greater than 95%, we only need the coefficients based on SSP5-8.5 scenario that can represent most spatial variations of gridded precipitation under SSP1-2.6.

[Figure]

**Figure R5** The comparison of average coefficients of the top 10 PCs in January under SSP5-8.5 (left column) and SSP1-2.6 (right column) after adjusting the order. The order of each column represents the PCs from top one to ten.

[Figure]

**Figure R5** Continued.

*(24) L283-292: More discussion of the issue of how much warming in your training affects the fit during the experimental phase, which is good to see. You point out that using a historical/cool training for a warmer scenario would be unwise; it might be better to state the opposite in acknowledgement of how you already pointed out that the 8.5 training generally produces worse results as you go from the 7.0 experiment to the 2.6 experiment. The last sentence here also is potentially very instructive for if/when you include an LCM-PREMU example, i.e. you would train PREMU based on the SSP whose future temperature most closely resembles the future temperatures from the LCM simulation in question.*

**[Response]** Thank you for this suggestion. We have added a short mention of it as the last sentence of this paragraph as an instructive suggestion for future PREMU implications: "When coupling PREMU with other LCMs to emulate the gridded precipitation under a new prescribed emission scenario, we suggest calibrating PREMU against the SSP whose future temperature most closely resembles the temperature in LCMs under that new scenario".

*(25) L304-305: While this is one possible conclusion to draw from the ESM results being more robust versus lag than the historical results were, you did previously posit "an alternative argument" suggesting that apparently better results in ESMs might be due to underrepresentation of topography and aerosols. As these are potential sources of variation, the "robustness" versus lag with ESMs could also be due to the ESMs not showing enough variation for the lag differences to matter.*

    **[Response]** Yes, we agree this is the possible reason for the robustness to lag-length in the fit to ESMs. Thus, we have added this in the revised version: "This may due to underrepresenting of topography and aerosol effects in ESMs, which are potential sources of variations in precipitation and could be important for influencing the lag-differences.".

*(26) L309-310: This could also be a potential concern since SSTs have a much larger influence on total atmospheric water vapor than land temperatures do, both by being much more surface area and by being the main source of evaporation. With that consideration, it should follow that PREMU should be at least slightly worse in its precipitation simulation – potentially still very good, but still not as good as when ocean area is also considered. I would like to see the differences between the normal emulator and emulator-land explicitly shown in some manner. You do state that including the ocean is still the preferred method for now; it would be good to expound on that.*

    **[Response]** Thank you, we agree, and so we have now added the comparison of Pearson correlation coefficients, side-by-side, for PREMU and PREMU-land. In the revised manuscript, we now write: "We found that the emulator-land can also reproduce consistent trends and interannual variations of GLAP and changes in gridded precipitation with ESMs (Fig. S25 and S26), while the correlation in interannual variations of GLAP with ESMs is relatively lower than PREMU (R = 0.71 for SSP1-2.6; R = 0.88 for SSP2-4.5; R = 0.91 for SSP3-7.0; R = 0.96 for SSP5-8.5).".

*(27) Table 3: You have space at the upper left for a proper title and/or stating the units; the latter in particular should be easily read somewhere outside of just the caption.*

    **[Response]** Yes, the title and units have been added.

*(28) Figure 4: I would love to see difference plots e-a, e-c, f-b, and f-d.*

    **[Response]** Thanks for the suggestion. Actually, we only provide a rough range rather than exact value. We would like to show the comparison of changes in precipitation in Fig. 4, so perhaps an easier way to represent the performance of PREMU. The difference plots have been shown in Fig. R6 and Fig. R7 (also shown as Fig. S5 and Fig. S6) and now writing:

"The overestimation of MAP with our PREMU method is mainly found in the Tibetan Plateau and central Africa (~20%), and the underestimation of MAP mainly in northern Siberia, Greenland Island and northern Australia (-15% – -30%; Fig. 3c and Fig. S5)",and: "In some regions of East Asia, Europe, Australia and South America, PREMU underestimates / overestimates the changes in annual precipitation (in range 50-200 mm year-1; Fig. S6)".

[Figure]

**Figure R6 a**) The differences in MAP between GSWP3 and emulations from PREMU. **b**) The differences in MAP between emulations from PREMU and Simple Linear method.

[Figure]

**Figure R7 a**) The differences in changes of MAP between GSWP3 and emulations from PREMU. **b**) The differences in changes of MAP between emulations from PREMU and Simple Linear method.

*(29) Figure 5 onward: From here on out, your maps are smaller than earlier; between the small size and the fact that the first red+blue from the middle are similar to the grey, all of the maps here and onward are hard to read. I would strive to rearrange all of these figures so that you are no more than two maps wide per page – and potentially darken those first lighter colors a bit as well.*

**[Response]** Following the suggestion, we have revised the colors in Fig. 5 and Fig. 7 and the corresponding figures in SI. We darkened the lighter colors and lightened the gray background color. We have also enlarged the size of Figure 5 and onward (although noting for the journal, that the width of the image cannot exceed 19cm). As for Figure 10, we prefer to leave it unchanged, otherwise the height of this figure would be more than a full side if we are to keep only two maps wide per page. We are open about whether rearranging this figure as no more than two maps wide.

*(30) Figure 7: In trying to make this figure no more than two maps wide, try visualizing it as a 2x2 grid where each quadrant has three maps stacked vertically.*

**[Response]** Thanks for the suggestion. After consideration, we deleted the middle column of this figure which shows the MAP of emulation from PREMU. Then the figure satisfies no more than two maps wide without loss of information.

*(31) Figure 8: I understand that 8.5 is the training scenario, then followed by 2.6 to 7.0 in that order as the experiments, but it still looks odd that the overall order isn't uniformly increasing or decreasing.*

**[Response]** To keep consistency with Figure 5, we would like to keep the order of Figure 8. However, for avoiding misleading, we added the text "Calibration" and "Verification" in Figure 7 and Figure 8, to the left side of figure.

**Technical Corrections:**

*(32) There seems to be a minor inconsistency in the paper between whether 2015 or 2016 is the starting point for the ESM data, L125 vs L132. Please make sure these are better aligned.*

**[Response]** Thanks. We have revised all starting point for ESM data as 2015 for both manuscript and SI.

*(33) L117: remove "the" form "discussed in the Sect. 4"*

**[Response]** It has been revised.

**References**

Efthymiadis, D., Jones, P. D., Briffa, K. R., Böhm, R., and Maugeri, M.: Influence of large-scale atmospheric circulation on climate variability in the Greater Alpine Region of Europe, J. Geophys. Res.: Atmospheres, 112, https://doi.org/10.1029/2006JD008021, 2007.

Li, W., Zhai, P., and Cai, J.: Research on the Relationship of ENSO and the Frequency of Extreme

Precipitation Events in China, Adv. Clim. Change. Res., 2, 101-107, https://doi.org/10.3724/SP.J.1248.2011.00101, 2011.

Lu, B., Li, H., Wu, J., Zhang, T., Liu, J., Liu, B., Chen, Y., and Baishan, J.: Impact of El Niño and Southern Oscillation on the summer precipitation over Northwest China, Atmos. Sci. Lett, 20, e928, https://doi.org/10.1002/asl.928, 2019.

Jiang, Y., Cooley, D., and Wehner, M. F.: Principal Component Analysis for Extremes and Application to U.S. Precipitation, J. Climate. 33, 6441-6451, 10.1175/JCLI-D-19-0413.1, 2020.

Singh, C. V.: Pattern characteristics of Indian monsoon rainfall using principal component analysis (PCA), Atmos. Res., 79, 317-326, https://doi.org/10.1016/j.atmosres.2005.05.006, 2006.

Yan, Z., Wu, B., Li, T., Collins, M., Clark, R., Zhou, T., Murphy, J., and Tan, G.: Eastward shift and extension of ENSO-induced tropical precipitation anomalies under global warming, Sci. Adv., 6, eaax4177, 10.1126/sciadv.aax4177, 2020.

---

## Referee Report (RR1)

Specific comments:

Section 5.1: The higher error in MAP for both PREMU and the linear regression seems to occur in Australia and West Africa (or more across the Sahel) which surround arid areas. It may also be that these areas have low relative changes in precipitation and hence prediction with principal components based on gridded Tair becomes more tough. This could be easily checked by the significance in their coefficient values obtained for Equation 2.

Editorial comments:

R4.2

"Thus, Lower complexity models (LCMs) are designed as the common approaches to improve computational efficiency in climate change research **by focussing** on the most impact relevant variables"

R4.4

The emulators developed by Beusch and Mckinnon are a bit different, consider restructuring to e.g.

"Joint temperature and precipitation emulation by considering anthropogenic GHG forcing and large-scale modes of Sea Surface Temperature (SST) variability has proven possible (Mckinon and Deser 2018; Mckinon and Deser 2021). More recently, a spatially resolved emulator (MESMER) solely requiring GMT e.g. by coupling to the emission-driven LCM (MAGICC), to then generate annual temperature fields, has been developed (Beusch et al. 2021)"

R4.5

 "Precipitation has high spatio-temporal variability and is affected by atmospheric dynamics **and** inter-annual modes of variability (Li et al., 2021; Tsanis and Tapoglou, 2019), making **represention of it** with traditional LCM approaches difficult. Thus, only two LCMs (IMOGEN and OSCAR) have tried to emulate precipitation**, but with poor skill** (Zelazowski et al., 2018; Gasser et al., 2017). IMOGEN emulates the gridded precipitation based on the regression relationship (by month and location) between gridded precipitation and global land average temperature (Zelazowski et al., 2018). OSCAR constructs the emulator by establishing **a** relationship between global average precipitation and global average temperature and radiative forcing, **from which a pattern-scaling method is used to deduce the gridded precipitation** (Gasser et al., 2017). Nevertheless, the gridded precipitation estimated by the simple linear method is not **fully** reliable **in either** IMOGEN **or** OSCAR; the gridded precipitation predicted by IMOGEN  explains less than 20% of variance of seasonal precipitation in most regions (Zelazowski et al., 2018) and OSCAR cannot capture …

L204: Refer to ONeill  (https://doi.org/10.5194/gmd-9-3461-2016)
when referencing SSP 5-8.5, consider also rephrasing :

due to it representing the most extreme changes in Tair amongst SSPs.

L209 and L318: Set out in  Table

L214: Do you have examples of the "limited area application", it is not too important but a reference here may help give an idea about what improvements your study's approach brings.

L215: and hence  … has a k dependency.

L312:  where

L315: Do you mean?:

we constructed 315 the regression relationship between GLAP and the 10 individual principal components  separately.

Section 3.2.2 seems to be more of a "Generating emulations using PREMU" than "Validation". Section 3.3 seems to capture validation quite well so maybe consider renaming?

L320-L323: Based on the principal component coefficients extracted using these calibration datasets, we then estimated the $T_{e,m}^{PCA}(y,t)$ using Equation 1, for 1951- 2016 using Tair from GSWP3 and for 2015-2100 using Tair from each ESM independently under the other three SSPs .

L403: The percentage error at each grid **point**

L459: A key requirement of **PREMU** is that it…

L485: West Africa

L486: which **is** discussed

L591: considering restructuring the sentence "…, that are account for in ESM simulations,…" seems to be floating without any subject.

L607: the SSP5-8.5 scenario and SSP1-2.6 scenario. Though with a different order of PCA coefficients (Fig. S19-S20), this suggests…

L623: . There are some studies that predict  an increased variability in precipitation under a warmer world  (Zhang et al., 2021; Song et al., 2018),

L635: Throughout this study, we used  the three-month average temperature…

L650: This may **be** due to…

---

## Author Response (AR2)

**Author's Response to the Review Comments**

We appreciate the time and effort by the Editor and the reviewer in assessing our manuscript and the thoughtful and valuable suggestions offered. We are pleased that our reviewer advises now only "minor" revisions, but we have still taken every suggestion seriously and responded in full as described below. We hope that our paper amendments are to the satisfaction of the reviewer and that the manuscript can be considered for publication in GMD.

In the following reply, the black italic text is the reviewer's requests, the black indented text is our responses and the blue text in quotes are revised text in the manuscript.

**1. Response to Comments from Editor:**

*Are the links and DOI to code & data up-to-date? Is it easy enough for someone to reproduce your results and build upon your work? etc.*

**[Response]** Thank you for reminding us of this. We have uploaded the latest code to GitHub. In addition, we also uploaded the driving data so that it is publicly available, with all links shown in the README file in GitHub. Finally, we use the GitHub-to-Zenodo integration and updated the related DOI in the manuscript. With these data and code, readers can reproduce our results in full following the instructions provided.

**2. Response to Comments from Reviewer #1:**

*Section 5.1: The higher error in MAP for both PREMU and the linear regression seems to occur in Australia and West Africa (or more across the Sahel) which surround arid areas. It may also be that these areas have low relative changes in precipitation and hence prediction with principal components based on gridded Tair becomes more tough. This could be easily checked by the significance in their coefficient values obtained for Equation 2.*

**[Response]** Thank you for this suggestion. We agree that the higher error in changes of MAP for PREMU and the linear regression occurs in the arid areas (Fig. 4**b, d, f**). Following the reviewer's suggestion, we have checked the significance of the correlation between the precipitation from GSWP3 and the estimation from PREMU (as given by Eq.2) during the period 1901-1950 for each month (Figs. R1-R2). We find that there is almost no difference in significance level in the correlation between these arid areas and other regions. However, notable is that the coefficient of variation of precipitation (shown as the ratio of the standard deviation to the mean) is much greater in arid areas (e.g., Australia and Sahel; Fig. R3-4 below). Hence these two factors together imply that the regression models in these regions still fit well variations as well as trends - yet for arid regions we can expect both PREMU and linear regression to show larger absolute errors in MAP.

[Figure]

**Figure R1** The correlation coefficients between precipitation from the GSWP3 dataset and estimates from PREMU. Statistic is calculated for each month (a-l respectively), and for the period 1901-1950.

[Figure]

**Figure R2** The significance of correlation between precipitation from the GSWP3 dataset and estimates from PREMU. Statistic is calculated for each month (a-l respectively), and for the period 1901-1950.

[Figure]

**Figure R3** The ratio of the standard deviation in precipitation to the mean of precipitation, using the GSWP3 dataset. Statistic is calculated for each month (a-l respectively), and for the period 1901-1950.

[Figure]

**Figure R4** The average ratio (across all months) of the standard deviation to the mean of the GSWP3-based values presented in Fig. R3.

**Editorial Comments:**

*(1) R4.2 "Thus, Lower complexity models (LCMs) are designed as the common approaches to improve computational efficiency in climate change research **by focussing** on the most impact relevant variables".*

    **[Response]** We have revised the wording, following the reviewer's suggestion.

*(2) R4.4 The emulators developed by Beusch and Mckinnon are a bit different, consider restructuring to e.g.*

*"Joint temperature and precipitation emulation by considering anthropogenic GHG forcing and large-scale modes of Sea Surface Temperature (SST) variability has proven possible (Mckinon and Deser 2018; Mckinon and Deser 2021). More recently, a spatially resolved emulator (MESMER) solely requiring GMT e.g. by coupling to the emission-driven LCM (MAGICC), to then generate annual temperature fields, has been developed (Beusch et al. 2021)".*

    **[Response]** Thank you. The paper has been revised, using the suggested wording.

*(3) R4.5 "Precipitation has high spatio-temporal variability and is affected by atmospheric dynamics **and** inter-annual modes of variability (Li et al., 2021; Tsanis and Tapoglou, 2019), making **represention of it** with traditional LCM approaches difficult. Thus, only two LCMs (IMOGEN and OSCAR) have tried to emulate precipitation**, but***

**with poor skill** *(Zelazowski et al., 2018; Gasser et al., 2017). IMOGEN emulates the gridded precipitation based on the regression relationship (by month and location) between gridded precipitation and global land average temperature (Zelazowski et al., 2018). OSCAR constructs the emulator by establishing **a** relationship between global average precipitation and global average temperature and radiative forcing, **from which a pattern-scaling method is used to deduce the gridded precipitation** (Gasser et al., 2017). Nevertheless, the gridded precipitation estimated by the simple linear method is not **fully** reliable **in either** IMOGEN **or** OSCAR; the gridded precipitation predicted by IMOGEN  explains less than 20% of variance of seasonal precipitation in most regions (Zelazowski et al., 2018) and OSCAR cannot capture…".*

**[Response]** Again, thank you for providing a suggested reword, that we have adopted in our paper.

*(4) L204: Refer to ONeill (https://doi.org/10.5194/gmd-9-3461-2016) when referencing SSP 5-8.5, consider also rephrasing :*
*due to it representing the most extreme changes in Tair amongst SSPs.*

**[Response]** Thank you. This reference has been added and, and the related text revised as: "For the estimates of future change, we used precipitation and Tair from SSP5-8.5 scenario of greenhouse gas increases to calibrate the ESM-specific emulator. We selected this scenario due to it representing the most extreme changes in Tair amongst SSPs. (O'Neill et al., 2016).".

*(5) L209 and L318: Set out in  Table*
**[Response]** It has been revised.

*(6) L214: Do you have examples of the "limited area application", it is not too important but a reference here may help give an idea about what improvements your study's approach brings.*
**[Response]** This reference has been added (Rahaman et al., 2019): "However, unlike many limited-area applications that derive a single timeseries (to multiply each PCA; Rahaman et al., 2019)".

*(7) L215: and hence  … has a k dependency*
**[Response]** It has been revised.

*(8) L312:  where*
**[Response]** It has been revised.

*(9) L315: Do you mean?:*
*we constructed 315 the regression relationship between GLAP and the 10 individual principal components  separately.*

**[Response]** Thank you. Yes, we do mean that, and the paper has been revised accordingly.

*(10) Section 3.2.2 seems to be more of a "Generating emulations using PREMU" than "Validation". Section 3.3 seems to capture validation quite well so maybe consider renaming?*
    **[Response]** Thanks for the suggestion. We have renamed Section 3.2.2 as "Generating emulations using PREMU" and Section 3.3 as "Validation".

*(11) L320-L323: Based on the principal component coefficients extracted using these calibration datasets, we then estimated the $T_{e,m}^{PCA}(y,i)$ using Equation 1, for 1951-2016 using Tair from GSWP3 and for 2015-2100 using Tair from each ESM independently under the other three SSPs*
    **[Response]** It has been revised.

*(11) L403: The percentage error at each grid **point***
    **[Response]** It has been revised, as suggested.

*(11) L459: A key requirement of **PREMU** is that it···*
    **[Response]** It has been revised.

*(11) L485: West Africa*
    **[Response]** Thank you. This geographical description has been corrected throughout the text.

*(11) L486: which **is** discussed*
    **[Response]** It has been revised.

*(11) L591: considering restructuring the sentence "···, that are account for in ESM simulations,···" seems to be floating without any subject*
    **[Response]** Thanks. After consideration, we decided to delete this and revised as: "We speculate that this may be caused by the most drastic land use changes associated with that former SSP scenario, resulting in a slower increase in precipitation under SSP3-7.0 (Riahi et al., 2017).".

*(11) L607: the SSP5-8.5 scenario and SSP1-2.6 scenario. Though with a different order of PCA coefficients (Fig. S19-S20), this suggests···*
    **[Response]** It has been revised, as suggested.

*(11) L623: . There are some studies that predict  an increased variability in precipitation under a warmer world (Zhang et al., 2021; Song et al., 2018),*
    **[Response]** Thanks. It has been revised.

*(11) L635: Throughout this study, we used  the three-month average temperature⋯*

    **[Response]** It has been revised.

*(11) L650: This may **be** due to⋯*
**[Response]** It has been revised.